# Purifying Shampoo: Investigating Shampoo's Heuristics by Decomposing its Preconditioner

**Runa Eschenhagen**[*,1]    **Aaron Defazio**[2]    **Tsung-Hsien Lee**[†]
**Richard E. Turner**[1,3]    **Hao-Jun Michael Shi**[4]

[1]Department of Engineering, University of Cambridge
[2]Fundamental AI Research, Meta Superintelligence Labs, Meta Platforms, Inc.
[3]The Alan Turing Institute
[4]Infrastructure Optimizations, Meta Superintelligence Labs, Meta Platforms, Inc.

## Abstract

The recent success of Shampoo in the AlgoPerf contest has sparked renewed interest in Kronecker-factorization-based optimization algorithms for training neural networks. Despite its success, Shampoo relies heavily on several heuristics such as learning rate grafting and stale preconditioning to achieve performance at-scale. These heuristics increase algorithmic complexity, necessitate further hyperparameter tuning, and lack theoretical justification. This paper investigates these heuristics from the angle of Frobenius norm approximation to full-matrix Adam and decouples the preconditioner's eigenvalues and eigenbasis updates. We show that grafting from Adam mitigates the staleness and mis-scaling of the preconditioner's *eigenvalues* and how correcting the eigenvalues directly eliminates the need for learning rate grafting. To manage the error induced by infrequent *eigenbasis* computations, we propose an adaptive criterion for determining the eigenbasis computation frequency motivated by terminating a warm-started QR algorithm. This criterion decouples the update frequency of different preconditioner matrices and enables us to investigate the impact of approximation error on convergence. These practical techniques offer a principled angle towards removing Shampoo's heuristics and developing improved Kronecker-factorization-based training algorithms.

## 1  Introduction

Structured non-diagonal, and especially Kronecker-factored, preconditioned stochastic gradient algorithms have been extensively studied for neural network training (Heskes, 2000; Martens, 2010; Martens & Grosse, 2015; Li, 2018; Gupta et al., 2018). Despite their promise, diagonally preconditioned methods like Adam have remained the de facto methods for training neural networks over the past decade (Duchi et al., 2011; Kingma & Ba, 2015). Recently, a distributed implementation of the Shampoo algorithm (Gupta et al., 2018; Anil et al., 2020; Shi et al., 2023) won the external tuning track of the AlgoPerf neural network training algorithm competition (Dahl et al., 2023; Kasimbeg et al., 2025).[1] This result has renewed interest in non-diagonally preconditioned training algorithms, inspiring methods like Muon (Jordan et al., 2024; Bernstein, 2025) and SOAP (Vyas et al., 2025a).

---

Correspondence to: `re393@cam.ac.uk` and `hjmshi@meta.com`.

[*]Work performed while an intern and external research collaborator at FAIR, Meta Platforms.

[†]Work performed while employed by AI and Systems Co-Design, Meta Platforms.

[1]The *external tuning* track requires a submission to specify a hyperparameter search space, in contrast to the hyperparameter-tuning-free *self-tuning* track.

39th Conference on Neural Information Processing Systems (NeurIPS 2025).

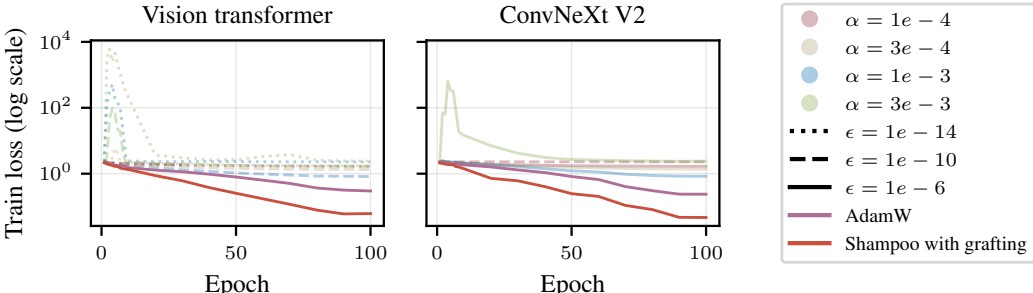

Figure 1: Shampoo with stale preconditioner (updating the root inverse matrices every $F = 100$ steps) without grafting for different choices of the learning rate $\alpha$ and $\epsilon$ on Imagewoof. All tested hyperparameter combinations underperform AdamW and, by extension, Shampoo with Adam grafting.

However, the winning Shampoo submission to the AlgoPerf competition relied on several crucial heuristics beyond Shampoo (Anil et al., 2020; Shi et al., 2023). Most notably, each layer's update is re-scaled by the update magnitude of a reference optimizer, a technique known as *learning rate grafting* (Agarwal et al., 2020); see Figure 1 for an illustration of its importance. Additionally, to reduce its computational overhead, the root-inverse of the preconditioner is only re-computed every 100 steps, resulting in the use of a *stale* preconditioner. Despite their empirical effectiveness, both heuristics lack theoretical justification and remain poorly understood.

In this paper, we investigate the role of these two heuristics by decoupling the updates of the preconditioner's eigenvalues and eigenbasis. In Section 3, we empirically demonstrate that Shampoo requires learning rate grafting in order to address the staleness and mis-scaling of the preconditioner's *eigenvalues*. Correcting Shampoo's eigenvalues at every step like in SOAP (Vyas et al., 2025a) removes the need for grafting. We further formalize this intuition by comparing bounds on the update magnitude of Shampoo and full-matrix Adam.

Given the importance of controlling the approximation error of the eigenvalues, we next consider the frequency of the *eigenbasis* updates in Section 4. Motivated by a termination criterion for the QR algorithm that bounds the relative error of the Kronecker factor approximation induced by the current eigenbasis (Golub & Van Loan, 2013), we propose an adaptive method for determining the eigenbasis update frequency. Our empirical results show that the approximation error of the Kronecker factors evolves during training and impacts convergence, depending on both the training stage and parameter's properties. Using an adaptive update frequency can improve Shampoo's training efficiency, especially when more frequent eigenbasis computations accelerate convergence.

## 2 Background

We consider neural network training as a standard stochastic optimization problem, where the goal is to minimize the expected loss function

$$\min_{\boldsymbol{\theta} \in \mathbb{R}^d} \mathcal{L}(\boldsymbol{\theta}) = \mathbb{E}_{(\boldsymbol{x},\boldsymbol{y}) \sim p_{\mathcal{D}}(\boldsymbol{x},\boldsymbol{y})}\big[\ell(f_{\boldsymbol{\theta}}(\boldsymbol{x}), \boldsymbol{y})\big], \tag{1}$$

where $f_{\boldsymbol{\theta}} : \mathbb{R}^N \to \mathbb{R}^c$ is the neural network prediction function with parameters $\boldsymbol{\theta} \in \mathbb{R}^d$ (flattened and concatenated into a vector), $p_{\mathcal{D}}(\boldsymbol{x}, \boldsymbol{y})$ is the joint data distribution from which inputs $\boldsymbol{x} \in \mathbb{R}^N$ and targets $\boldsymbol{y} \in \mathbb{R}^c$ are sampled, and $\ell : \mathbb{R}^c \times \mathbb{R}^c \to \mathbb{R}$ is the loss function. The neural network is commonly trained using preconditioned stochastic gradient methods that update the parameters at each iteration $t$ by

$$\boldsymbol{\theta}_{t+1} = \boldsymbol{\theta}_t - \alpha_t \boldsymbol{C}_t^{-p} \boldsymbol{g}_t = \boldsymbol{\theta}_t - \alpha_t \boldsymbol{Q}_{\boldsymbol{C}_t} \boldsymbol{\Lambda}_{\boldsymbol{C}_t}^{-p} \boldsymbol{Q}_{\boldsymbol{C}_t}^{\mathsf{T}} \boldsymbol{g}_t, \tag{2}$$

where $\boldsymbol{g}_t = \nabla_{\boldsymbol{\theta}_t} \ell(f_{\boldsymbol{\theta}_t}(\boldsymbol{x}_t), \boldsymbol{y}_t) \in \mathbb{R}^d$ is the stochastic (mini-batch) gradient with respect to the sample $(\boldsymbol{x}_t, \boldsymbol{y}_t) \sim p_{\mathcal{D}}(\boldsymbol{x}, \boldsymbol{y})$, $\alpha_t > 0$ is the step size, $p > 0$ is the exponent, and $\boldsymbol{C}_t \in \mathbb{R}^{d \times d}$ is a symmetric positive-definite *preconditioner* or *scaling* matrix. The second equality in Equation (2) expresses the update using the eigendecomposition of the preconditioner matrix $\boldsymbol{C}_t = \boldsymbol{Q}_{\boldsymbol{C}_t} \boldsymbol{\Lambda}_{\boldsymbol{C}_t} \boldsymbol{Q}_{\boldsymbol{C}_t}^{\mathsf{T}}$, where the eigenbasis matrix $\boldsymbol{Q}_t \in \mathbb{R}^{d \times d}$ is orthogonal and eigenvalue matrix $\boldsymbol{\Lambda}_{\boldsymbol{C}_t} \in \mathbb{R}^{d \times d}$ is diagonal.

We primarily focus on $C_t$ as an accumulation of gradient outer products, i.e., $\bar{A}_t = \sum_{s=1}^{t} g_s g_s^{\mathsf{T}}$ or $A_t = \beta_2 A_{t-1} + (1 - \beta_2) g_t g_t^{\mathsf{T}}$ with $p = 1/2$, which correspond to full-matrix AdaGrad and RMSprop/Adam, respectively (Duchi et al., 2011; Kingma & Ba, 2015), although this can be generalized to other alternatives, like the natural gradient method (Amari, 1998); see Appendix A. When $C_t$ is diagonal, this scaling recovers simplified versions of popular algorithms like AdaGrad (Duchi et al., 2011) or Adam (Kingma & Ba, 2015), ignoring additional modifications like exponential moving averages, momentum, bias corrections, etc. However, for non-diagonal $C_t$, these methods require storing and possibly inverting dense $d \times d$ matrices, which is computationally prohibitive.

To address this, two structured approximations are commonly applied: (1) a layer-wise block-diagonal approximation, where each block captures pairwise correlations within each layer; (2) a Kronecker-factored approximation that takes a $mn \times mn$ matrix block for each layer and approximates it as a Kronecker product of two smaller matrices of size $m \times m$ and $n \times n$. The Kronecker product approximation is particularly convenient for computing the inverse-root matrix-vector product, leading to the design of algorithms like Kronecker-Factored Approximate Curvature (K-FAC) (Heskes, 2000; Martens & Grosse, 2015; Grosse & Martens, 2016; Martens et al., 2018; Eschenhagen et al., 2023), Shampoo (Gupta et al., 2018; Anil et al., 2020; Shi et al., 2023), and their variants (George et al., 2018; Gao et al., 2020; Ren & Goldfarb, 2021; Feinberg et al., 2023; Duvvuri et al., 2024; Lin et al., 2024b,a). Alternative approaches, such as Hessian-free or inexact Newton methods, avoid explicitly storing the preconditioner matrix, e.g., by leveraging Hessian-vector products (Martens, 2010; Li, 2018; Pooladzandi & Li, 2024).

## 2.1 Shampoo

The Shampoo preconditioner was originally developed as a Kronecker-factorized upper bound to full-matrix AdaGrad in its regret analysis (Gupta et al., 2018). For a weight matrix $W_t \in \mathbb{R}^{m \times n}$ with stochastic gradient $G_t \in \mathbb{R}^{m \times n}$ where $g_t = \text{vec}(G_t)$, Shampoo stores symmetric positive semi-definite matrices that approximate an *idealized* preconditioner with $L_t \approx \mathbb{E}[G_t G_t^{\mathsf{T}}] \in \mathbb{R}^{m \times m}$ and $R_t \approx \mathbb{E}[G_t^{\mathsf{T}} G_t] \in \mathbb{R}^{n \times n}$, and updates the preconditioner and weight matrix by

$$L_t = \beta_2 L_{t-1} + (1 - \beta_2) G_t G_t^{\mathsf{T}}, \tag{3}$$

$$R_t = \beta_2 R_{t-1} + (1 - \beta_2) G_t^{\mathsf{T}} G_t, \tag{4}$$

$$W_{t+1} = W_t - \alpha_t L_t^{-\frac{1}{4}} G_t R_t^{-\frac{1}{4}} \tag{5}$$

given $\beta_2 \in [0, 1)$.[2][3] The original Shampoo algorithm uses a sum to accumulate the gradient outer products, although the exponential moving average is more commonly used in practice. Therefore, we are interested in approximating the preconditioner of full-matrix Adam, given by $A_t = \beta_2 A_{t-1} + (1 - \beta_2) g_t g_t^{\mathsf{T}} \approx \mathbb{E}[g_t g_t^{\mathsf{T}}]$.

The search direction or update in matrix form is given as $U_t^{\text{Shampoo}} = -L_t^{-\frac{1}{4}} G_t R^{-\frac{1}{4}}$ or, equivalently, $C_t^{\text{Shampoo}} = (R_t \otimes L_t)^{\frac{1}{2}}$ with $p = 1/2$ in Equation (2). We also consider *Shampoo*[2] defined as $C_t^{\text{Shampoo}^2} = R_t \otimes L_t$, which is a tighter approximation to full-matrix AdaGrad (Morwani et al., 2025). Both updates can be generalized to higher-order tensors.

## 2.2 Eigenvalue-corrected Shampoo and SOAP

By leveraging a Kronecker product approximation, Shampoo's preconditioner is restricted to a Kronecker product structure for both its eigenvectors and eigenvalues. Specifically, if we have a preconditioner $C_t = R_t \otimes L_t$ with eigendecompositions $L_t = Q_{L_t} \Lambda_{L_t} Q_{L_t}^{\mathsf{T}}$ and $R_t = Q_{R_t} \Lambda_{R_t} Q_{R_t}^{\mathsf{T}}$, then $C_t$ has eigendecomposition $C_t = (Q_{R_t} \otimes Q_{L_t})(\Lambda_{R_t} \otimes \Lambda_{L_t})(Q_{R_t} \otimes Q_{L_t})^{\mathsf{T}}$. Preconditioning $-g_t$ with $C_t^{\text{Shampoo}}$ and $p = 1/2$ in its matrix form is equivalent to the matrix transformation:

$$U_t^{\text{Shampoo}} = -L_t^{-\frac{1}{4}} G_t R_t^{-\frac{1}{4}} = -Q_{L_t} \Lambda_{L_t}^{-\frac{1}{4}} (Q_{L_t}^{\mathsf{T}} G_t Q_{R_t}) \Lambda_{R_t}^{-\frac{1}{4}} Q_{R_t}^{\mathsf{T}}. \tag{6}$$

---

[2]We drop the subscript in the expectation, taken with respect to $(x_t, y_t) \sim p_{\mathcal{D}}(x, y)$.

[3]A pseudo-inverse that ignores the null space of the matrix or perturbing the matrix by a regularization term $\epsilon I$ for $\epsilon > 0$ can be used to handle the symmetric positive semi-definite case.

This can be interpreted as applying a Kronecker-factored coordinate-wise scaling to a gradient with changed basis, i.e., $\tilde{\boldsymbol{G}}_t = \boldsymbol{Q}_{\boldsymbol{L}_t}^\mathsf{T} \boldsymbol{G}_t \boldsymbol{Q}_{\boldsymbol{R}_t}$, scaling the transformed gradient, and converting it back to its original basis.

Instead of restricting the eigenvalues to be a Kronecker product, Liu et al. (2018) and George et al. (2018) have proposed *correcting* the eigenvalues by decoupling the scaling from the basis in K-FAC. This involves computing a separate scaling matrix $\boldsymbol{D}_t \approx \mathbb{E}\big[\tilde{\boldsymbol{G}}_t^{\odot 2}\big] \in \mathbb{R}^{m \times n}$ and using it in place of the preconditioner's original eigenvalues $\mathrm{mat\,diag}(\boldsymbol{\Lambda}_{\boldsymbol{R}_t} \otimes \boldsymbol{\Lambda}_{\boldsymbol{L}_t})$.[4][5]

Anil et al. (2020) noted that this correction can also be applied to Shampoo. Most recently, Vyas et al. (2025a) presented promising empirical results for language models using an instance of eigenvalue-corrected Shampoo called SOAP, which updates the scaling by $\boldsymbol{D}_t = \beta_2 \boldsymbol{D}_{t-1} + (1 - \beta_2)\tilde{\boldsymbol{G}}_t^{\odot 2}$. Since then, SOAP has also been shown to perform well for physics-informed neural networks (Wang et al., 2025) and to reduce outlier features in transformers, which potentially improves quantization (He et al., 2024). We refer to our practical instantiation of eigenvalue-corrected Shampoo as *EShampoo* (Appendix B, Algorithm 2), which uses the same $\boldsymbol{D}_t$ as SOAP. See Appendix D.1 for clarification on the distinction between eigenvalue correction, EShampoo, and SOAP.

## 3 Learning rate grafting compensates for Shampoo's eigenvalues

Originally motivated by decoupling an optimizer's update magnitude from its direction to account for different implicit learning rate schedules, Agarwal et al. (2020) proposed *learning rate grafting*, a technique that combines the layer-wise update of one optimizer with the layer-wise update magnitude of another. For Shampoo, this means taking Shampoo's layer-wise update $\boldsymbol{U}_t^{\mathrm{Shampoo}}$ and rescaling it by the Frobenius norm of the grafting method's update $\boldsymbol{U}_t^{\mathrm{Grafting}}$ (typically Adam):

$$\boldsymbol{W}_{t+1} = \boldsymbol{W}_t + \alpha_t \frac{||\boldsymbol{U}_t^{\mathrm{Grafting}}||_F}{||\boldsymbol{U}_t^{\mathrm{Shampoo}}||_F} \, \boldsymbol{U}_t^{\mathrm{Shampoo}}. \tag{7}$$

A complete description of Shampoo with Adam grafting is given in Appendix B, Algorithm 3.

This approach has been critical to Shampoo's empirical success (Anil et al., 2020; Shi et al., 2023; Kasimbeg et al., 2025). As shown in Figure 1, Shampoo without grafting (and updating the root inverse matrices every $F = 100$ steps) underperforms AdamW, with many hyperparameter settings diverging. Anil et al. (2020) suggests that grafting is used to account for differences in magnitude of the eigenspectrum of the Kronecker factors for different layers, as well as the infrequent updates of the Kronecker factors and their inverse roots (see Anil et al. (2020), Appendix G). However, these claims have not been thoroughly investigated.

To focus our empirical investigation, we study Shampoo with Adam grafting and $F = 100$, which was the winning configuration that was used in the external tuning track of the AlgoPerf competition (Kasimbeg et al., 2025). Since grafting re-scales the layer-wise update based on its magnitude, we analyze the Frobenius norm of the updates of full-matrix and diagonal Adam, Shampoo, and EShampoo. The magnitude of the eigendecomposed update in Equation (2) with Kronecker-factored $\boldsymbol{Q}_{\boldsymbol{C}_t} = \boldsymbol{Q}_{\boldsymbol{R}_t} \otimes \boldsymbol{Q}_{\boldsymbol{L}_t}$ is determined by the norm of the stochastic gradient $\boldsymbol{G}_t$ and the eigenvalues of $\boldsymbol{C}_t$:

**Lemma 1.** *Let* $\boldsymbol{U} = \boldsymbol{Q}_{\boldsymbol{L}}(\boldsymbol{D}^{\odot -p} \odot (\boldsymbol{Q}_{\boldsymbol{L}}^\mathsf{T} \boldsymbol{G} \boldsymbol{Q}_{\boldsymbol{R}}))\boldsymbol{Q}_{\boldsymbol{R}}^\mathsf{T} \in \mathbb{R}^{m \times n}$ *be the generalized eigendecomposed Kronecker-factored update given by orthogonal matrices* $\boldsymbol{Q}_{\boldsymbol{L}} \in \mathbb{R}^{m \times m}$, $\boldsymbol{Q}_{\boldsymbol{R}} \in \mathbb{R}^{n \times n}$, *and dense scaling matrix* $\boldsymbol{D} \in \mathbb{R}^{m \times n}$, *with* $p > 0$. *Then we have:*

$$(\max_{i,j} \boldsymbol{D}_{i,j})^{-p}||\boldsymbol{G}||_F \leq ||\boldsymbol{U}||_F \leq (\min_{i,j} \boldsymbol{D}_{i,j})^{-p}||\boldsymbol{G}||_F. \tag{8}$$

Lemma 1 covers multiple algorithms. We can recover the idealized Adam update by setting $\boldsymbol{Q}_{\boldsymbol{L}} = \boldsymbol{I} \in \mathbb{R}^{m \times m}$, $\boldsymbol{Q}_{\boldsymbol{R}} = \boldsymbol{I} \in \mathbb{R}^{n \times n}$, and $\boldsymbol{D} = \mathbb{E}\big[\boldsymbol{G}^{\odot 2}\big]$ with $p = 1/2$. The idealized Shampoo update can be recovered through the choice of $\boldsymbol{D} = \mathrm{mat\,diag}(\boldsymbol{\Lambda}_{\boldsymbol{R}} \otimes \boldsymbol{\Lambda}_{\boldsymbol{L}})$ with $p = 1/4$, $\boldsymbol{L} = \mathbb{E}[\boldsymbol{G}\boldsymbol{G}^\mathsf{T}] = \boldsymbol{Q}_{\boldsymbol{L}}\boldsymbol{\Lambda}_{\boldsymbol{L}}\boldsymbol{Q}_{\boldsymbol{L}}^\mathsf{T}$ and $\boldsymbol{R} = \mathbb{E}[\boldsymbol{G}^\mathsf{T}\boldsymbol{G}] = \boldsymbol{Q}_{\boldsymbol{R}}\boldsymbol{\Lambda}_{\boldsymbol{R}}\boldsymbol{Q}_{\boldsymbol{R}}^\mathsf{T}$ (or $p = 1/2$ for Shampoo[2]). Idealized EShampoo is recovered with the choice of $\boldsymbol{D} = \mathbb{E}\big[(\boldsymbol{Q}_{\boldsymbol{L}}^\mathsf{T} \boldsymbol{G} \boldsymbol{Q}_{\boldsymbol{R}})^{\odot 2}\big]$ and $p = 1/2$ instead.

---

[4] $\boldsymbol{X}^{\odot 2}$ denotes the element-wise square.

[5] $\mathrm{mat\,diag}(\cdot)$ takes a $mn \times mn$ matrix and reshapes its diagonal into a $m \times n$ matrix.

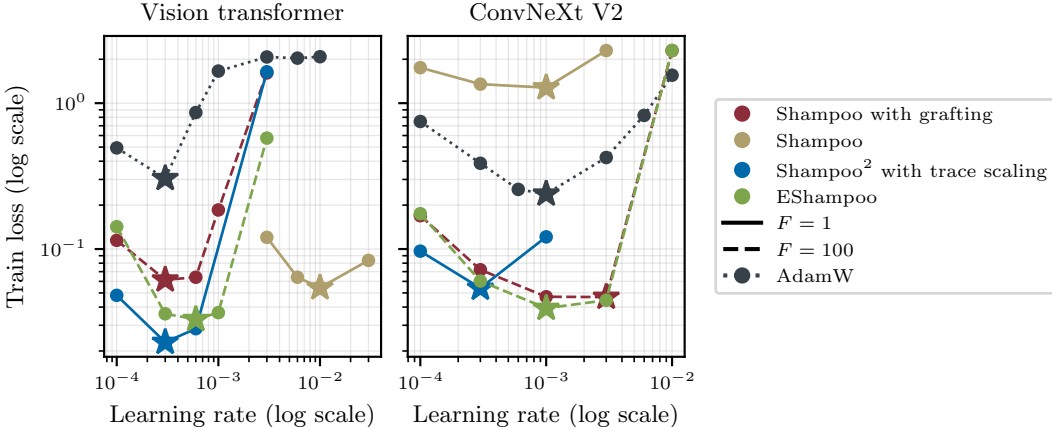

Figure 2: Training results with different Shampoo variants and eigendecomposition frequencies $F$ on the Imagewoof dataset. Shampoo with eigenvalue correction achieves a better training loss compared to Shampoo with Adam grafting, and the optimal learning rate for Adam transfers to both variants.

**Proposition 1.** *Assume that $\mathbb{E}[\boldsymbol{gg}^\top]$ is symmetric positive definite. The magnitude of the idealized updates for full-matrix Adam, diagonal Adam, and EShampoo are all bounded by the power of the extreme eigenvalues of full-matrix Adam:*

$$\lambda_{\max}(\mathbb{E}[\boldsymbol{gg}^\top])^{-p}\|\boldsymbol{G}\|_F \leq \|\boldsymbol{U}\|_F \leq \lambda_{\min}(\mathbb{E}[\boldsymbol{gg}^\top])^{-p}\|\boldsymbol{G}\|_F, \tag{9}$$

*for all $p > 0$. However, under the simplifying assumption that $\mathbb{E}[\boldsymbol{G}] = \boldsymbol{0}$ and $\boldsymbol{G}_{i,j}$ is independent from $\boldsymbol{G}_{k,l}$ for $(i, j) \neq (k, l)$ and has bounded second moment, $\lambda_{\min}(\mathbb{E}[\boldsymbol{gg}^\top]) \leq \mathbb{E}[\boldsymbol{G}_{i,j}^2] \leq \lambda_{\max}(\mathbb{E}[\boldsymbol{gg}^\top])$ and Shampoo has dimension-dependent bounds:*

$$m^{-p/2}n^{-p/2}\lambda_{\max}(\mathbb{E}[\boldsymbol{gg}^\top])^{-p}\|\boldsymbol{G}\|_F \leq \|\boldsymbol{U}\|_F \leq m^{-p/2}n^{-p/2}\lambda_{\min}(\mathbb{E}[\boldsymbol{gg}^\top])^{-p}\|\boldsymbol{G}\|_F. \tag{10}$$

See Appendix C for the proofs of Lemma 1 and Proposition 1.

This highlights a key issue: Shampoo's update magnitude can be mis-scaled relative to Adam and EShampoo, especially due to dimension-dependent factors. The basis does not influence the update magnitude – only the eigenvalues do. While this additional scaling can be absorbed into the learning rate when only handling a single matrix, it is potentially problematic when one needs to handle multiple parameters simultaneously. In addition, the use of stale eigenvalues can result in update magnitudes that lie outside of the bounds of full-matrix Adam. This leads us to the following hypothesis:

> **The role of learning rate grafting in Shampoo**
>
> Learning rate grafting compensates for the scaling and staleness of Shampoo's eigenvalues.

From the Frobenius norm approximation perspective, using Shampoo$^2$ (via $\boldsymbol{C}_t^{\text{Shampoo}^2}$) yields a tighter approximation to full-matrix Adam compared to $\boldsymbol{C}_t^{\text{Shampoo}}$ (Morwani et al., 2025), which addresses the mismatch of the eigenvalues' exponent in Equation (10). Additionally, rescaling the preconditioner by $S^{-1} = \text{Tr}(\boldsymbol{R}_t)^{-1} = \text{Tr}(\boldsymbol{L}_t)^{-1}$ ensures exactness when full-matrix Adam $\boldsymbol{A}_t$ is a Kronecker product (Morwani et al., 2025). The scaling $S^{-1}$ has also been previously introduced for Tensor Normal Training (Ren & Goldfarb, 2021, TNT) to approximate the Fisher information matrix.

Based on our observations, we can make several predictions:

1. With $F = 1$, Shampoo without grafting should perform well when layer scalings are similar, but may struggle with highly variable parameter shapes.

2. Using $S^{-1}\boldsymbol{C}_t^{\text{Shampoo}^2}$ with $F = 1$ should address scaling and staleness and match Shampoo with grafting. This is exact when full-matrix Adam decomposes into a Kronecker product.

3. Updating an eigenvalue correction at every iteration (e.g. like in SOAP) should also address scaling and staleness, matching Shampoo with grafting.

**Empirical validation.** We ablate different variants of Shampoo on the Imagewoof dataset with vision transformer (ViT) and ConvNeXt V2 models. We plot the final training loss after 100 epochs against the learning rate. Following the specification of the AlgoPerf Shampoo submission, we update the preconditioner every 100 steps when grafting the learning rate from Adam, and the eigenbasis when using eigenvalue correction. Our results are presented in Figure 2. We observe that all three predictions are confirmed: Shampoo$^2$ scaled by $S^{-1}$ or eigenvalue correction is able to match or even surpass Shampoo with grafting on both problems, and Shampoo without grafting can match the final loss for the Imagewoof ViT workload, but not for the ConvNeXt V2 model.

**Hyperparameter transfer.** The optimal learning rate of Adam transfers to Shampoo with grafting and EShampoo. However, it is not apparent whether the optimal learning rate transfers to Shampoo$^2$ scaled by $S^{-1}$. Learning rate transfer is only sensible when jointly transferring $\epsilon$ since it affects the effective step size, but Shampoo does not directly add $\epsilon$ to the root of the eigenvalues, as done in Adam and other diagonal-scaling-based optimizers. While we have not tested this, the optimal learning rate may potentially transfer when matching the effective $\epsilon$.

**Computational and memory costs.** While updating Shampoo every iteration incurs a prohibitive computational overhead, the eigenvalue correction is only slightly more expensive than Adam grafting since it requires more matrix products. EShampoo adds the same memory overhead to Shampoo as Adam grafting, specifically requiring an additional $d$-dimensional buffer for the second moment accumulation. One could reduce this overhead by adopting techniques such as Adam-mini, which only requires a single scalar per parameter block, defined according to the Hessian structure (Zhang et al., 2025b), and could be applied to both grafting and the eigenvalue correction. Alternatively, Liu et al. (2025) propose to scale Muon to approximately match the RMS norm of Adam's update to enable the re-use of Adam's hyperparameters, which may provide a practical heuristic that removes grafting without any memory overhead. Finally, an eigenvalue correction for the individual Kronecker factors can also correct for the staleness of their eigenvalues and has, in fact, been shown to also remove the need for grafting, while reducing the necessary buffer size from $mn$ to $m + n$ for each $m \times n$ weight matrix (Lin et al., 2025).

**Applicability to other methods.** The issue of stale eigenvalues extends to all sparsely-updated Kronecker-factored preconditioners such as K-FAC variants and TNT. Although we are unaware of any work that combines K-FAC or TNT with grafting, it is common practice to apply the Adam update to gradients preconditioned with K-FAC (Pauloski et al., 2021; Osawa et al., 2023a,b; Eschenhagen et al., 2023), which we hypothesize could serve a similar function as grafting.

## 4 Controlling the approximation error induced by stale eigenbases

While frequent updates to the preconditioner's eigenvalues are important, the impact of periodically computing the eigenbasis remains less clear. The optimal frequency of the eigenbasis computation should be chosen to balance the time of each iteration against the method's per-iteration convergence rate, and may depend on the stage of training, layer type, and the specific Kronecker factors ($L_t$, $R_t$) involved. Shampoo currently uses a fixed update frequency for all matrices, which ignores these distinctions and the computational cost associated with different parameter shapes.

### 4.1 Can we adaptively control the approximation error induced by the stale eigenbasis?

In SOAP, the initial eigenbasis is computed via an eigendecomposition during the first iteration, then is refined by a single power iteration and QR decomposition at all subsequent iterations, c.f. Algorithm 4 in Vyas et al. (2025a), attributed to Wang et al. (2024).[6] The method corresponds to a single iteration of a warm-started simultaneous iteration, an extension of the power iteration for iteratively computing eigendecompositions (Golub & Van Loan, 2013). However, this approach may still yield a poor approximation to the true eigenbasis, especially when the eigenbasis changes rapidly due to the choice of $\beta_2$ or the dynamics of the stochastic gradient.

To address this, we propose controlling the error induced by the eigenbasis using a *warm-started QR algorithm* that iterates until a relative approximation error criterion is satisfied. This approach offers three major benefits: (1) each Kronecker factor can be treated independently, allowing adaptive frequency across different layers, factors, and stages of training; (2) evaluating the criterion with the

---

[6]This approach was referred to as randomized SVD in Wang et al. (2024), Appendix B.

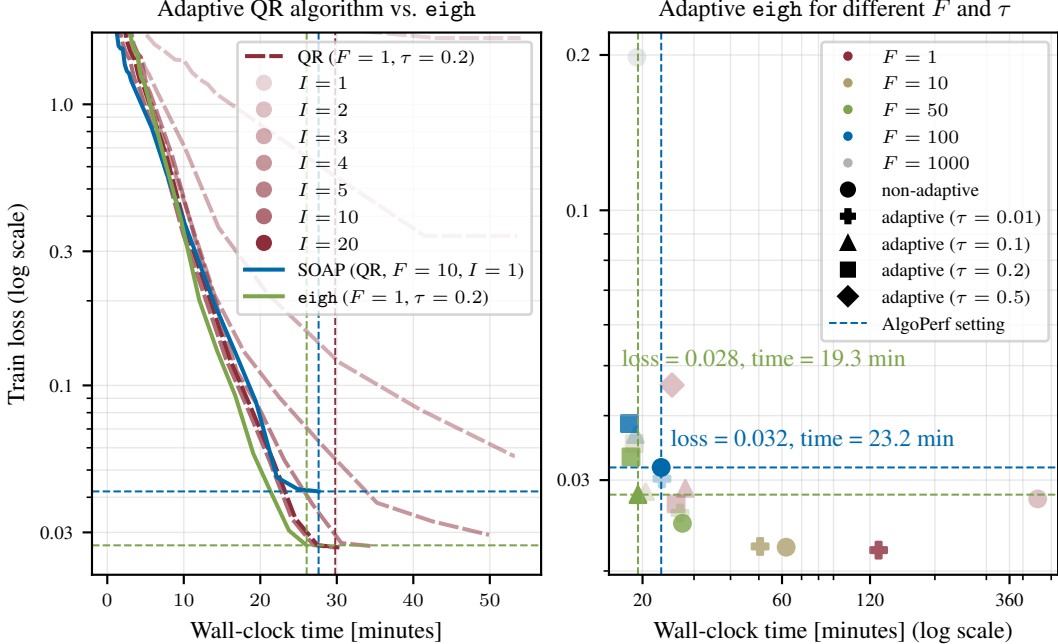

Figure 3: All configurations are for EShampoo. (**left**) On the Imagewoof ViT problem, setting the maximum number of iterations $I < 10$ with threshold $\tau = 0.2$ for the adaptive QR algorithm leads to significant increase in wall-clock time compared to using adaptive `eigh`. Even with $I = 10$, adaptive `eigh` is faster. The default SOAP setting achieves worse final loss and is also slightly slower. (**right**) Using the adaptive criterion to determine when to skip the eigendecomposition (`eigh`) improves efficiency by 20% in wall-clock time compared to updating every 100 iterations (AlgoPerf setting).

last computed eigenbasis prior to the first QR iteration enables us to *skip* eigenbasis updates; and (3) the error in the eigenbasis can be controlled through a threshold $\tau \in [0, 1)$, quantifying *inexactness*.

Consider a single Kronecker factor $\boldsymbol{L}_t \in \mathbb{R}^{m \times m}$ without loss of generality. We want to solve inexactly for an orthogonal matrix $\boldsymbol{Q}_{\boldsymbol{L}_t}$ such that we obtain an approximation to the eigendecomposition $\boldsymbol{L}_t = \boldsymbol{Q}_{\boldsymbol{L}_t} \boldsymbol{\Lambda}_{\boldsymbol{L}_t} \boldsymbol{Q}_{\boldsymbol{L}_t}^{\mathsf{T}}$, with the diagonal eigenvalues matrix $\boldsymbol{\Lambda}_{\boldsymbol{L}_t}$. If the previous approximate eigenbasis $\hat{\boldsymbol{Q}}_{\boldsymbol{L}_{t-1}}$ is a good approximation to the current eigenbasis $\boldsymbol{Q}_{\boldsymbol{L}_t}$, then the approximate eigenvalues induced by the stale eigenbasis $\hat{\boldsymbol{\Lambda}}_{\boldsymbol{L}_t} = \hat{\boldsymbol{Q}}_{\boldsymbol{L}_{t-1}}^{\mathsf{T}} \boldsymbol{L}_t \hat{\boldsymbol{Q}}_{\boldsymbol{L}_{t-1}}$ should be nearly diagonal.

We can leverage this observation to define a unified criterion for skipping the eigenbasis computation and terminating the warm-started QR algorithm. Diagonalizing $\hat{\boldsymbol{\Lambda}}_{\boldsymbol{L}_t}$ gives an approximation of $\boldsymbol{L}_t$, specifically, $\hat{\boldsymbol{L}}_t = \hat{\boldsymbol{Q}}_{\boldsymbol{L}_{t-1}} \operatorname{diag}(\hat{\boldsymbol{\Lambda}}_{\boldsymbol{L}_t}) \hat{\boldsymbol{Q}}_{\boldsymbol{L}_{t-1}}^{\mathsf{T}}.$[7] We can bound the relative error of this approximation induced by the stale eigenbasis, or equivalently the relative error in the Frobenius norm of the off-diagonal entries of $\hat{\boldsymbol{\Lambda}}_{\boldsymbol{L}_t}$, which is cheaper to compute:

> **Adaptive eigenbasis computation frequency criterion**
>
> $$\frac{||\boldsymbol{L}_t - \hat{\boldsymbol{L}}_t||_F}{||\boldsymbol{L}_t||_F} = \frac{||\hat{\boldsymbol{Q}}_{\boldsymbol{L}_{t-1}}^{\mathsf{T}} \boldsymbol{L}_t \hat{\boldsymbol{Q}}_{\boldsymbol{L}_{t-1}} - \operatorname{diag}(\hat{\boldsymbol{\Lambda}}_{\boldsymbol{L}_t})||_F}{||\hat{\boldsymbol{Q}}_{\boldsymbol{L}_{t-1}}^{\mathsf{T}} \boldsymbol{L}_t \hat{\boldsymbol{Q}}_{\boldsymbol{L}_{t-1}}||_F} = \frac{||\hat{\boldsymbol{\Lambda}}_{\boldsymbol{L}_t} - \operatorname{diag}(\hat{\boldsymbol{\Lambda}}_{\boldsymbol{L}_t})||_F}{||\hat{\boldsymbol{\Lambda}}_{\boldsymbol{L}_t}||_F} \le \tau. \quad (11)$$

This condition provides a guarantee of the quality of the eigendecomposition approximation. If the condition is satisfied, we reuse the previous eigenbasis $\hat{\boldsymbol{Q}}_{\boldsymbol{L}_t} = \hat{\boldsymbol{Q}}_{\boldsymbol{L}_{t-1}}$; otherwise, we refine it through the QR algorithm until the criterion is met. A complete pseudocode is provided in Appendix B, Algorithm 4. Note that evaluating the approximate eigenvalues $\hat{\boldsymbol{\Lambda}}_{\boldsymbol{L}_t}$ at the first step requires two matrix multiplications, which is significantly cheaper than computing a step of the QR algorithm.

---

[7]$\operatorname{diag}(\cdot)$ takes a vector/matrix and returns a diagonal matrix with the vector/matrix's diagonal on its diagonal.

Table 1: Results on a subset of the AlgoPerf workloads. We show the mean and standard error of the steps/time to the targets across the runs that reach them. See Appendix E, Table 3 for more results.

| Workload | Shampoo Variant | Hits Target | Steps | Time [min] |
|---|---|---|---|---|
| FastMRI | Adam grafting ($F = 100$) | 4/5 | $4301 \pm 109$ | $13.96 \pm 0.44$ |
| | $C^{\text{EShampoo}}$ ($F = 100$) | 5/5 | $2536 \pm 66$ | $10.44 \pm 0.21$ |
| | $C^{\text{EShampoo}}$ ($\tau = 0.1, F = 50$) | 5/5 | $2468 \pm 145$ | $10.81 \pm 0.72$ |
| ImageNet ViT | Adam grafting ($F = 100$) | 1/1 | 79907 | 894.27 |
| | $C^{\text{EShampoo}}$ ($F = 100$) | 1/1 | 76226 | 894.85 |
| | $C^{\text{EShampoo}}$ ($\tau = 0.1, F = 50$) | 1/1 | 77459 | 935.89 |
| OGBG | Adam grafting ($F = 100$) | 2/5 | $12574 \pm 708$ | $39.20 \pm 1.88$ |
| | $C^{\text{EShampoo}}$ ($F = 100$) | 3/5 | $8320 \pm 1203$ | $33.02 \pm 4.05$ |
| | $C^{\text{EShampoo}}$ ($\tau = 0.1, F = 50$) | 5/5 | $7117 \pm 328$ | $27.55 \pm 3.49$ |

## 4.2 Does this adaptivity translate to efficiency gains?

The practical efficiency gains of adaptively controlling the approximation error depend on multiple factors. First, the efficiency of the QR algorithm is determined by how its computational cost compares to a standard eigendecomposition, which in turn depends on both the cost of each QR iteration and the total number of iterations needed to satisfy the threshold $\tau$. Second, evaluating the criterion adds a small constant overhead independent of the actual frequency of eigendecompositions.

Interestingly, we found that setting a smaller maximum number of QR iterations ($I < 10$) per step results in a significant slowdown in wall-clock time, as the total number of QR iterations accumulated over all training steps is higher than when using a larger maximum ($I \geq 10$). However, even with a larger number of maximum iterations, the QR algorithm was slightly more expensive than computing eigendecompositions with `torch.linalg.eigh` (shortened as `eigh`) whenever Equation (11) does not hold. The default configuration of SOAP, which computes a single step of the warm-started simultaneous iteration every 10 steps, was also slightly slower in wall-clock time compared to adaptively computing eigendecompositions and reaches a worse final loss. This result conflicts with the observation in Figure 7 in Vyas et al. (2025a), which may be due to differences in the types of workloads considered; see Figure 3 (left).

Because of this result, we primarily leverage the criterion for determining the frequency of calling `eigh`. To further bound the total possible number of eigendecompositions performed over the course of training, we evaluate the criterion at a fixed frequency $F$ as opposed to every step. We ablate different choices of $F$ and $\tau$ in Figure 3 (right). We compare against the baseline setting of the winning AlgoPerf submission, which re-computes the eigendecomposition of all factor matrices every 100 steps on the Imagewoof ViT problem. We observe that this setting without adaptivity requires 20% more wall-clock time compared to an adaptive setting ($\tau = 0.1, F = 50$).

To test whether the results generalize to other problems, we consider a subset of the AlgoPerf workloads and compare the winning Shampoo submission with Adam grafting to: (1) EShampoo with the same fixed eigenbasis computation frequency of $F = 100$; and (2) EShampoo with $\tau = 0.1$ and $F = 50$; see Table 1. First, as predicted by Section 3, EShampoo with the fixed eigenbasis update frequency matches or outperforms Shampoo with Adam grafting in steps and wall-clock time, using the same hyperparameters. Second, using the adaptive criterion with $\tau = 0.1$ and $F = 50$ matches the fixed frequency $F = 100$ for the FastMRI and OGBG workloads, and does slightly worse for ImageNet ViT. We also present results with other fixed and adaptive frequencies in Appendix E, Table 3. Overall, we find that the performance on the considered problems is quite robust to the eigenbasis computation frequency (c.f. Appendix E, Table 3). We expect that when Shampoo's convergence benefits from more frequent eigenbasis updates, adaptively determining the frequency can result in higher efficiency.

## 4.3 What patterns of adaptivity emerge?

The criterion in Equation (11) also provides insight into how rapidly the eigenbasis changes for different parameter types. On Imagewoof ViT, we set $\tau = 0.1$ and check the criterion at every

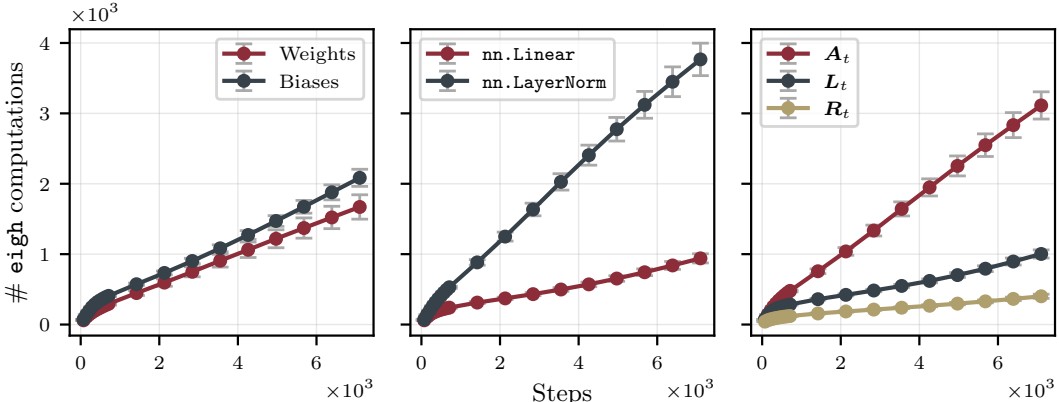

Figure 4: We show the mean with standard error across preconditioners corresponding to the labels in the legends, for EShampoo with $\tau = 0.01$ and $F = 1$ on Imagewoof ViT. The eigenbases for biases and layer normalization parameters are changing faster than for weight matrices and linear layers, respectively.

step ($F = 1$); see Figure 4. We observe more frequent eigendecomposition computations early in training that trend towards a constant update frequency at the end of training. When comparing different parameter types, we find that the eigenbases of preconditioners for bias terms evolve more rapidly than those for weights. Similarly, layer normalization parameters require almost $4\times$ as many eigenbasis updates as linear layer parameters.

We also compare the number of eigendecompositions across the two Kronecker factors $\boldsymbol{L}_t$ and $\boldsymbol{R}_t$ for weight matrices, and $\boldsymbol{A}_t$ for biases and layer normalization parameters.[8] The eigenbases of $\boldsymbol{A}_t$ are updated most frequently, followed by $\boldsymbol{L}_t$, then $\boldsymbol{R}_t$. The same trend holds when removing learnable layer norm parameters and for ConvNeXt V2 trained on the same dataset (c.f. Appendix E, Figure 7). A similar analysis for a Llama 3 model with 324 million parameters trained on 3.2 billion tokens of C4 data reveals a different trend, namely, $\boldsymbol{L}_t$ evolves faster than $\boldsymbol{R}_t$ and $\boldsymbol{A}_t$, which correspond to RMS normalization parameters (c.f. Appendix E, Figure 8).

### 4.4 How does the error induced by the eigenbasis affect convergence?

While we have focused on controlling the error induced by the preconditioner's stale eigenbasis, our primary concern is its effect on convergence, rather than the error itself. In the Imagewoof ViT setting, we find that the approximation quality (determined by $\tau$) during early iterations is far more critical than during later iterations; see Figure 5 (left). Specifically, setting $\tau = 0.8$ for the first 90% and $\tau = 0.01$ for the last 10% of iterations yields a final loss nearly identical to using $\tau = 0.8$ throughout all of training ($\approx 0.09 - 0.1$). In contrast, setting $\tau = 0.01$ for the first 10% and $\tau = 0.8$ for the remaining 90% of iterations leads to a significantly better final loss ($\approx 0.04$).

Remarkably, freezing the eigenbasis after the first iteration (by setting $\tau = 0.99$ for all of training) still significantly outperforms AdamW ($\approx 0.12$ vs. $0.3$).[9] This highlights that frequent eigenbasis computations are most beneficial early in training, consistent with previous observations on Shampoo, PSGD, and SOAP (Ishikawa & Yokota, 2024; Walters et al., 2025; Nestler, 2025; Vyas et al., 2025b).

To further investigate the discrepancy in the rate of change in the eigenbases corresponding to 1D and 2D parameters (c.f. Figure 4, right), we compare an Imagewoof ViT run with $\tau = 0.01$ for 2D parameters and no change of basis, i.e. Adam, for 1D parameters and vice versa. Surprisingly, using Adam for 1D parameters does not affect Shampoo's convergence or final loss. Conversely, running Adam for 2D parameters and changing the basis according to $\tau = 0.01$ for 1D parameters closely matches Adam's convergence and final loss; see Figure 5 (right).

---

[8]Full-matrix Adam is also used for weight matrices $\boldsymbol{W}_t \in \mathbb{R}^{m \times n}$ with $mn \leq$ `max_preconditioner_dim`, which is a hyperparameter to control the largest possible dimension of the preconditioner.

[9]For 5 out of 172 Kronecker factors, the eigenbasis is computed twice instead of just once.

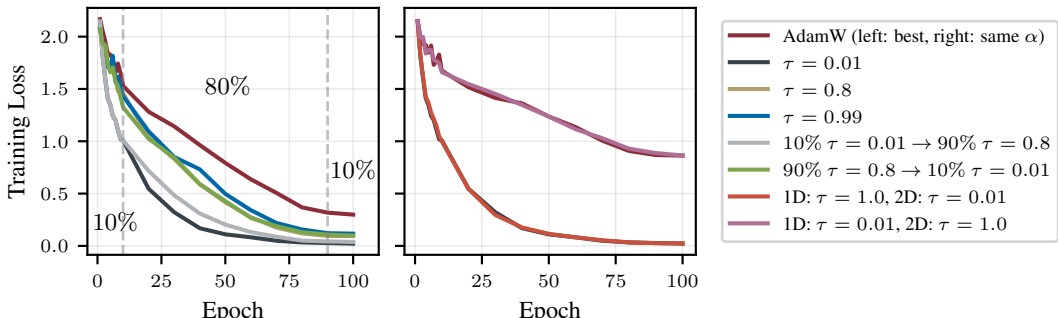

Figure 5: All configurations are for EShampoo. (**left**) The error in the eigenbases is dramatically more important for early iterations. A single eigenbasis computation at the first iteration ($\tau = 0.99$) is sufficient to outperform AdamW. (**right**) The difference between the convergence behavior of AdamW and EShampoo on this problem can be exclusively attributed to the eigenbases corresponding to 2D parameters.

Given that $45\%$ of the preconditioner matrices correspond to 1D parameters and their eigenbases change more rapidly, this suggests that many eigendecompositions are not needed at all. For example, using $\tau = 0.01$ for all parameters increases runtime by $2.9\times$ compared to only applying EShampoo to 2D parameters and Adam to 1D parameters, with no improvement in final loss. In practice, SOAP already uses Adam for 1D parameters in order to reduce its computational and memory overhead (Vyas et al., 2025a).

## 5 Discussion and conclusion

In this paper, we demonstrate that frequently updating the eigenvalues while periodically updating the eigenbasis of Shampoo's preconditioner provides a principled and practical approach for eliminating grafting. It remains an open question whether the same correction applies directly to the AdaGrad summation, or if a more sophisticated, basis-aware eigenvalue correction is needed (c.f. Appendix D.2). We also show how controlling the approximation error induced by the stale eigenbasis can improve efficiency. In order to determine the eigenbasis computation frequency in a truly problem-agnostic manner, we must understand how approximation error impacts convergence, as well as integrate systems-level considerations, such as batched kernel efficiency, into our algorithmic design. Further exploration is needed to understand how these trade-offs and techniques scale to larger models, such as large language models. While our work focuses on Shampoo, the ideas are not limited to the AdaGrad family, and can be adapted to other methods such as K-FAC and TNT.

From a theoretical perspective, a major open question is how to incorporate approximation quality into regret bounds for Shampoo (c.f. Appendix D.1). Finally, while we have implicitly treated full-matrix Adam as the right algorithm to approximate, alternative interpretations of Shampoo may better explain Shampoo's effectiveness in practice (Carlson et al., 2015a,b; Benzing, 2022; Bernstein & Newhouse, 2024; Maes et al., 2024; Pethick et al., 2025; Zhang et al., 2025a; Xie et al., 2025).

## Acknowledgments

We thank Anna Cai, Parameswaran Raman, and Ke Sang for their in-depth review of the paper. We are grateful for insightful discussions with Ganesh Ajjanagadde, Rohan Anil, Anna Cai, Rong Jin, Jihao Andreas Lin, Bruno Mlodozeniec, Vinay Rao, Isaac Reid, and Xingyu (Alice) Yang. We also acknowledge managerial support from Yuchen Hao, Guna Lakshminarayanan, Maxim Naumov, Sandeep Parab, Chunqiang Tang, and Lin Xiao. Lastly, we thank the anonymous reviewers for their productive feedback and suggestions of experiments and Kyunghun Nam and Yushun Zhang for pointing out minor errors in a preprint of this work.

Runa Eschenhagen is supported by ARM, the Cambridge Trust, and the Qualcomm Innovation Fellowship. Richard E. Turner is supported by Google, Amazon, ARM, Improbable and an EPSRC Prosperity Partnership (EP/T005386/1) and the EPSRC Probabilistic AI Hub (EP/Y028783/1).

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

Table 2: Shampoo variants and their properties.

| Shampoo variant | Justified from approximation perspective | Matches grafting in steps | Matches grafting in compute & memory | Learning rate transfer |
|---|---|---|---|---|
| grafting (Adam) | ✗ | ✓ | ✓ | ✓ |
| $C^{\text{Shampoo}}$ | ✗ | ✗ | ✗ | ✗ |
| $S^{-1}C^{\text{Shampoo}^2}$ | ✓ | ✓ | ✗ | ? |
| $C^{\text{EShampoo}}$ | ✓ | ✓ | ✓ | ✓ |

## A  Connecting full-matrix AdaGrad to the Fisher

We can also consider a more general class of preconditioned stochastic gradient methods with other choices of $C_t$ and $p > 0$ that update the parameters at each iteration $t$ by Equation (2). For example, one could choose $C_t$ as approximations of the Hessian $\nabla^2 \mathcal{L}(\boldsymbol{\theta})$ via subsampling or secant approximation with $p = 1$, which yields the class of *stochastic Newton* or *quasi-Newton* methods (Keskar & Berahas, 2016; Bollapragada et al., 2018; Berahas et al., 2020; Goldfarb et al., 2020). Another common choice is the Fisher information matrix $\boldsymbol{F}_t = \mathbb{E}_{\hat{\boldsymbol{y}}\sim f_{\boldsymbol{\theta}}(\boldsymbol{x}), \boldsymbol{x}\sim p_{\mathcal{D}}(\boldsymbol{x})}\big[\nabla_{\boldsymbol{\theta}}\ell(f_{\boldsymbol{\theta}}(\boldsymbol{x}),\hat{\boldsymbol{y}})\nabla_{\boldsymbol{\theta}}\ell(f_{\boldsymbol{\theta}}(\boldsymbol{x}),\hat{\boldsymbol{y}})^{\intercal}\big]$, which yields the *natural gradient* or, for common loss functions, *generalized Gauss-Newton method* with $p = 1$ (Amari, 1998; Martens, 2014). A summary of different methods is provided in Bottou et al. (2018).

By instead summing over per-sample gradient outer products, $\hat{\boldsymbol{A}}_t$ can be connected to the empirical Fisher. In general, the empirical Fisher is not expected to be a good approximation of the Fisher information matrix (Kunstner et al., 2019). However, by replacing the accumulation with an expectation over the conditional distribution given by the model $f_{\boldsymbol{\theta}}$, full-matrix AdaGrad can be made equivalent to the Fisher. This equivalence reveals a natural extension of Shampoo to approximate the Fisher, called Tensor Normal Training (Ren & Goldfarb, 2021, TNT), and is closely related to the K-FAC preconditioner (Anil et al., 2020). In fact, both approximations are exactly equivalent to the (block-diagonal) Fisher for simple cases such as deep linear networks (also with weight sharing) and mean squared error loss (Bernacchia et al., 2018; Eschenhagen et al., 2023; Morwani et al., 2025). The modifications in TNT deviate from the original Shampoo update (Gupta et al., 2018) because it was motivated by upper bounds to full-matrix AdaGrad in its non-smooth, convex regret analysis, rather than this Fisher approximation perspective.

## B  Algorithms pseudocode

In this section, we provide the pseudocode for all algorithms, including idealized (Algorithm 1) and practical (Algorithm 2) eigenvalue-corrected Shampoo, Shampoo with Adam grafting (Algorithm 3), and the adaptive warm-started QR algorithm (Algorithm 4). Algorithm 1 and Algorithm 3 present simplified versions of the algorithm ignoring common modifications like momentum or an exponential moving average over the stochastic gradient, bias corrections, and weight decay.

Note that different instances of eigenvalue-corrected Shampoo can be employed by changing the exponential moving average in Equation (12), eigenbasis computation in Equation (13), or eigenvalue correction update in Equation (14). SOAP delays the update of the eigenbasis until after the update step, approximates the eigenvalue correction (Equation (14)) by

$$\boldsymbol{D}_t = \beta_2 \boldsymbol{D}_{t-1} + (1 - \beta_2)\tilde{\boldsymbol{G}}_t^{\odot 2}, \tag{19}$$

and uses a single iteration of the warm-started simultaneous iteration. Additionally, there is a discrepancy between the SOAP algorithm presented in the paper and the official implementation: in the paper, the exponential moving average of the gradient $\boldsymbol{G}_t$ is used (line 4 of Algorithm 3 in Vyas et al. (2025a)), whereas the implementation computes the exponential moving average over the rotated gradient $\tilde{\boldsymbol{G}}_t$.[10] This can be interpreted as running Adam in Shampoo's eigenspace, which shares similarities to the GaLore algorithm (Zhao et al., 2024; Su et al., 2025).

---

[10]See https://github.com/nikhilvyas/SOAP/blob/f42d296cb4146a67fbe811371e6badb9a39cc54d/soap.py#L167.

---

**Algorithm 1** Idealized eigenvalue-corrected Shampoo pseudocode

---

**Require:** Parameter $\boldsymbol{W}_1 \in \mathbb{R}^{m\times n}$, learning rate $\alpha_t > 0$, $\epsilon > 0$.
1: Initialize $\boldsymbol{L}_0 = \boldsymbol{0} \in \mathbb{R}^{m\times m}$, $\boldsymbol{R}_0 = \boldsymbol{0} \in \mathbb{R}^{n\times n}$, and $\boldsymbol{D} = \boldsymbol{0} \in \mathbb{R}^{m\times n}$.
2: **for** $t = 1, ..., T$ **do**
3:     Compute (mini-batch) stochastic gradient: $\boldsymbol{G}_t = \nabla_\theta \ell(f_{\boldsymbol{\theta}_t}(\boldsymbol{x}), \boldsymbol{y})$.
4:     Update factor matrices:

$$\boldsymbol{L}_t = \beta_2 \boldsymbol{L}_{t-1} + (1-\beta_2)\boldsymbol{G}_t\boldsymbol{G}_t^\mathsf{T}, \quad \boldsymbol{R}_t = \beta_2 \boldsymbol{R}_{t-1} + (1-\beta_2)\boldsymbol{G}_t^\mathsf{T}\boldsymbol{G}_t. \tag{12}$$

5:     Compute orthonormal eigenbasis of the factor matrices:

$$\boldsymbol{Q}_{\boldsymbol{L}_t} = \mathrm{eigvec}(\boldsymbol{L}_t), \quad \boldsymbol{Q}_{\boldsymbol{R}_t} = \mathrm{eigvec}(\boldsymbol{R}_t). \tag{13}$$

6:     Transform gradient basis: $\tilde{\boldsymbol{G}}_t = \boldsymbol{Q}_{\boldsymbol{L}_t}^\mathsf{T}\boldsymbol{G}_t\boldsymbol{Q}_{\boldsymbol{R}_t}$.
7:     Compute or update eigenvalue correction:

$$\boldsymbol{D}_t^* = \underset{\boldsymbol{D}\in\mathbb{R}^{m\times n}}{\arg\min} \|\boldsymbol{A}_t - (\boldsymbol{Q}_{\boldsymbol{R}_t} \otimes \boldsymbol{Q}_{\boldsymbol{L}_t})\,\mathrm{diag}(\mathrm{vec}(\boldsymbol{D}))(\boldsymbol{Q}_{\boldsymbol{R}_t} \otimes \boldsymbol{Q}_{\boldsymbol{L}_t})\|_F. \tag{14}$$

8:     Compute update: $\boldsymbol{W}_{t+1} = \boldsymbol{W}_t - \alpha_t \boldsymbol{Q}_{\boldsymbol{L}_t}(\tilde{\boldsymbol{G}}_t \oslash (\sqrt{\boldsymbol{D}_t^*} + \epsilon \boldsymbol{1}\boldsymbol{1}^\mathsf{T}))\boldsymbol{Q}_{\boldsymbol{R}_t}$.
9: **end for**

---

 

---

**Algorithm 2** EShampoo pseudocode (as implemented for the experiments)

---

**Require:** Parameter $\boldsymbol{W}_1 \in \mathbb{R}^{m\times n}$, learning rate $\alpha_t > 0$, $\epsilon > 0, \beta_1, \beta_2 \in [0, 1)$, weight decay $\lambda \geq 0$, eigenbasis computation frequency $F \in \mathbb{N}$, and threshold $\tau \in [0, 1]$ for Equation (11) and Algorithm 4.
1: Initialize $\boldsymbol{M}_0 = \boldsymbol{0} \in \mathbb{R}^{m\times n}, \boldsymbol{L}_0 = \boldsymbol{0} \in \mathbb{R}^{m\times m}, \boldsymbol{R}_0 = \boldsymbol{0} \in \mathbb{R}^{n\times n}, \boldsymbol{Q}_{\boldsymbol{L}_0} = \boldsymbol{I}_m, \boldsymbol{Q}_{\boldsymbol{R}_0} = \boldsymbol{I}_n$, and $\boldsymbol{D}_0 = \boldsymbol{0} \in \mathbb{R}^{m\times n}$.
2: **for** $t = 1, ..., T$ **do**
3:     Compute (mini-batch) stochastic gradient: $\boldsymbol{G}_t = \nabla_\theta \ell(f_{\boldsymbol{\theta}_t}(\boldsymbol{x}), \boldsymbol{y})$.
4:     Compute exponential moving average of the gradient: $\boldsymbol{M}_t = \beta_1 \boldsymbol{M}_{t-1} + (1-\beta_1)\,\boldsymbol{G}_t$.
5:     Update factor matrices:

$$\boldsymbol{L}_t = \beta_2 \boldsymbol{L}_{t-1} + (1-\beta_2)\boldsymbol{G}_t\boldsymbol{G}_t^\mathsf{T}, \quad \boldsymbol{R}_t = \beta_2 \boldsymbol{R}_{t-1} + (1-\beta_2)\boldsymbol{G}_t^\mathsf{T}\boldsymbol{G}_t. \tag{15}$$

6:     **if** $t \mod F = 0$ and not Equation (11) **then**
7:         Compute eigenbasis of factor matrices (e.g. with `torch.linalg.eigh` or Algorithm 4):

$$\boldsymbol{Q}_{\boldsymbol{L}_t} = \mathrm{eigvec}(\boldsymbol{L}_t/(1-\beta_2^t)), \quad \boldsymbol{Q}_{\boldsymbol{R}_t} = \mathrm{eigvec}(\boldsymbol{R}_t/(1-\beta_2^t)). \tag{16}$$

8:     **else**

$$\boldsymbol{Q}_{\boldsymbol{L}_t} = \boldsymbol{Q}_{\boldsymbol{L}_{t-1}}, \quad \boldsymbol{Q}_{\boldsymbol{R}_t} = \boldsymbol{Q}_{\boldsymbol{R}_{t-1}}. \tag{17}$$

9:     **end if**
10:    Transform gradient basis: $\tilde{\boldsymbol{G}}_t = \boldsymbol{Q}_{\boldsymbol{L}_t}^\mathsf{T}\boldsymbol{G}_t\boldsymbol{Q}_{\boldsymbol{R}_t}$.
11:    Compute or update eigenvalue correction: $\boldsymbol{D}_t = \beta_2 \boldsymbol{D}_{t-1} + (1-\beta_2)\tilde{\boldsymbol{G}}_t^{\odot 2}$.
12:    Perform bias correction:

$$\tilde{\boldsymbol{M}}_t = \boldsymbol{M}_t/(1-\beta_1^t), \quad \tilde{\boldsymbol{D}}_t = \boldsymbol{D}_t/(1-\beta_2^t).$$

13:    Compute parameter update:

$$\boldsymbol{W}_{t+1} = \boldsymbol{W}_t - \alpha_t \left( \boldsymbol{Q}_{\boldsymbol{L}_t} \left( \boldsymbol{Q}_{\boldsymbol{L}_t}^\mathsf{T}\tilde{\boldsymbol{M}}_t\boldsymbol{Q}_{\boldsymbol{R}_t} \oslash \left( \sqrt{\tilde{\boldsymbol{D}}_t} + \epsilon \boldsymbol{1}\boldsymbol{1}^\mathsf{T} \right) \right) \boldsymbol{Q}_{\boldsymbol{R}_t}^\mathsf{T} + \lambda \boldsymbol{W}_t \right). \tag{18}$$

14: **end for**

---

---

**Algorithm 3** Shampoo with Adam grafting pseudocode

---

**Require:** Parameter $\boldsymbol{W}_1 \in \mathbb{R}^{m \times n}$, learning rate $\alpha_t > 0$, exponential moving average constant $\beta_2 \in (0, 1)$, $\epsilon > 0$.
1: Initialize $\boldsymbol{L}_0 = \boldsymbol{0} \in \mathbb{R}^{m \times m}$, $\boldsymbol{R}_0 = \boldsymbol{0} \in \mathbb{R}^{n \times n}$, and $\boldsymbol{D} = \boldsymbol{0} \in \mathbb{R}^{m \times n}$.
2: **for** $t = 1, ..., T$ **do**
3:      Compute (mini-batch) stochastic gradient: $\boldsymbol{G}_t = \nabla_\theta \ell(f_{\boldsymbol{\theta}_t}(\boldsymbol{x}), \boldsymbol{y})$.
4:      Update factor matrices:

$$\boldsymbol{L}_t = \beta_2 \boldsymbol{L}_{t-1} + (1 - \beta_2)\boldsymbol{G}_t \boldsymbol{G}_t^\mathsf{T}, \quad \boldsymbol{R}_t = \beta_2 \boldsymbol{R}_{t-1} + (1 - \beta_2)\boldsymbol{G}_t^\mathsf{T} \boldsymbol{G}_t.$$

5:      Update Adam grafting state: $\boldsymbol{D}_t = \beta_2 \boldsymbol{D}_{t-1} + (1 - \beta_2)\boldsymbol{G}_t^{\odot 2}$.
6:      Compute matrix root inverse of the factor matrices:

$$\boldsymbol{L}_t^{-1/4} = \mathrm{rootinv}(\boldsymbol{L}_t), \quad \boldsymbol{R}_t^{-1/4} = \mathrm{rootinv}(\boldsymbol{R}_t).$$

7:      Compute update: $\boldsymbol{W}_{t+1} = \boldsymbol{W}_t - \alpha_t \frac{\|-\boldsymbol{G}_t \oslash (\sqrt{\boldsymbol{D}_t} + \epsilon \boldsymbol{1}\boldsymbol{1}^T)\|_F}{\|-\boldsymbol{L}_t^{-1/4} \boldsymbol{G} \boldsymbol{R}_t^{-1/4}\|_F} \boldsymbol{L}_t^{-1/4} \boldsymbol{G}_t \boldsymbol{R}_t^{-1/4}$.
8: **end for**

---

---

**Algorithm 4** Warm-started QR iteration with relative error termination criterion.

---

**Require:** Matrix $\boldsymbol{L} = \boldsymbol{L}_t$, previous eigenbasis $\hat{\boldsymbol{Q}} = \hat{\boldsymbol{Q}}_{\boldsymbol{L}_{t-1}}$, relative tolerance $\tau \in [0, 1)$, maximum number of iterations $I$.
1: $i \leftarrow 0$
2: $\hat{\boldsymbol{\Lambda}} \leftarrow \hat{\boldsymbol{Q}}^\mathsf{T} \boldsymbol{L} \hat{\boldsymbol{Q}}$
3: **while** $\|\hat{\boldsymbol{\Lambda}} - \mathrm{diag}(\hat{\boldsymbol{\Lambda}})\|_F \leq \tau \|\hat{\boldsymbol{\Lambda}}\|_F$ and $i < I$ **do**
4:      $\boldsymbol{Q}, \boldsymbol{R} \leftarrow \texttt{QR}(\hat{\boldsymbol{\Lambda}})$
5:      $\hat{\boldsymbol{\Lambda}} \leftarrow \boldsymbol{R}\boldsymbol{Q}$
6:      $\hat{\boldsymbol{Q}} \leftarrow \hat{\boldsymbol{Q}}\boldsymbol{Q}$
7:      $i \leftarrow i + 1$
8: **end while**
9: **return** $\hat{\boldsymbol{Q}}, \hat{\boldsymbol{\Lambda}}$

---

In contrast, our implementation of EShampoo does not delay the update of the preconditioner, uses the same approximation of the eigenvalue correction as SOAP, uses `torch.linalg.eigh` for the eigendecomposition (unless indicated otherwise), and computes the exponential moving average over the gradient $\boldsymbol{G}_t$. Both algorithms use bias corrections and (decoupled) weight decay. We provide the pseudocode for EShampoo as it was implemented for our experiments in Algorithm 2. If $F \neq 1$, the algorithm reduces to AdamW until iteration $t = F$.

While we do not present experimental results here, Algorithm 4 or Equation (11) for `eigh` can also be used in Shampoo without an eigenvalue correction. This implementation stores the previous eigendecomposition of the Kronecker factors instead of the previous root-inverse Kronecker factors. When we skip an eigendecomposition, we maintain the previous eigenbasis and replace the previous eigenvalues with the estimated eigenvalues $\mathrm{diag}(\hat{\boldsymbol{\Lambda}}_{\boldsymbol{L}_t}) = \mathrm{diag}(\boldsymbol{Q}_{\boldsymbol{L}_{t-1}}^\mathsf{T} \boldsymbol{L}_t \boldsymbol{Q}_{\boldsymbol{L}_{t-1}})$ used in Equation (11). Alternatively, one can compute an exponential moving average over these estimated eigenvalues as done in KL-Shampoo (Lin et al., 2025), which also removes the need for grafting like the eigenvalue correction in SOAP and EShampoo (c.f. Section 3).

## C   Proofs

**Lemma 1.** *Let $\boldsymbol{U} = \boldsymbol{Q}_{\boldsymbol{L}}(\boldsymbol{D}^{\odot -p} \odot (\boldsymbol{Q}_{\boldsymbol{L}}^\mathsf{T} \boldsymbol{G} \boldsymbol{Q}_{\boldsymbol{R}}))\boldsymbol{Q}_{\boldsymbol{R}}^\mathsf{T} \in \mathbb{R}^{m \times n}$ be the generalized eigendecomposed Kronecker-factored update given by orthogonal matrices $\boldsymbol{Q}_{\boldsymbol{L}} \in \mathbb{R}^{m \times m}$, $\boldsymbol{Q}_{\boldsymbol{R}} \in \mathbb{R}^{n \times n}$, and dense scaling matrix $\boldsymbol{D} \in \mathbb{R}^{m \times n}$. Then we have:*

$$(\max_{i,j} \boldsymbol{D}_{i,j})^{-p} ||\boldsymbol{G}||_F \leq ||\boldsymbol{U}||_F \leq (\min_{i,j} \boldsymbol{D}_{i,j})^{-p} ||\boldsymbol{G}||_F. \tag{20}$$

*Proof.* Since the Frobenius norm of a matrix is invariant to orthogonal transformations and the entries of $D$ are bounded, i.e., $\min_{i,j} D_{i,j} \leq D_{i,j} \leq \max_{i,j} D_{i,j}$, we can show that

$$
\begin{aligned}
\|U\|_F &= \|Q_L(D^{\odot-p} \odot (Q_L^\mathsf{T} G Q_R))Q_R^\mathsf{T}\|_F \\
&= \|D^{\odot-p} \odot (Q_L^\mathsf{T} G Q_R))\|_F \\
&\in [(\max_{i,j} D_{i,j})^{-p}, (\min_{i,j} D_{i,j})^{-p}] \cdot \|Q_L^\mathsf{T} G Q_R\|_F \\
&\in [(\max_{i,j} D_{i,j})^{-p}, (\min_{i,j} D_{i,j})^{-p}] \cdot \|G\|_F.
\end{aligned}
$$

$\square$

**Proposition 1.** *Assume that $\mathbb{E}[gg^\mathsf{T}]$ is symmetric positive definite. The magnitude of the updates for full-matrix Adam, diagonal Adam, and eigenvalue-corrected Shampoo are all bounded by the power of the extreme eigenvalues of full-matrix Adam:*

$$
\lambda_{\max}(\mathbb{E}[gg^\mathsf{T}])^{-p}\|G\|_F \leq \|U\|_F \leq \lambda_{\min}(\mathbb{E}[gg^\mathsf{T}])^{-p}\|G\|_F, \tag{21}
$$

*for all $p > 0$. However, under the simplifying assumption that $\mathbb{E}[G] = 0$ and $G_{i,j}$ is independent from $G_{k,l}$ for $(i,j) \neq (k,l)$ and has bounded second moment, $\lambda_{\min}(\mathbb{E}[gg^\mathsf{T}]) \leq \mathbb{E}[G_{i,j}^2] \leq \lambda_{\max}(\mathbb{E}[gg^\mathsf{T}])$ and Shampoo has dimension-dependent bounds:*

$$
m^{-p/2}n^{-p/2}\lambda_{\max}(\mathbb{E}[gg^\mathsf{T}])^{-p}\|G\|_F \leq \|U\|_F \leq m^{-p/2}n^{-p/2}\lambda_{\min}(\mathbb{E}[gg^\mathsf{T}])^{-p}\|G\|_F. \tag{22}
$$

*Proof.* Note that Equation (21) holds for full-matrix Adam since $\|U\|_F = \|u\|_2 = \|\mathbb{E}[gg^\mathsf{T}]^{-p}g\|_2 \in [\lambda_{\max}(\mathbb{E}[gg^\mathsf{T}])^{-p}, \lambda_{\min}(\mathbb{E}[gg^\mathsf{T}])^{-p}] \cdot \|g\|_2$, where $u = \text{vec}(U)$ and $g = \text{vec}(G)$.

For diagonal Adam and eigenvalue-corrected Shampoo, it is sufficient to show that $\lambda_{\min}(\mathbb{E}[gg^\mathsf{T}]) \leq D_{i,j} \leq \lambda_{\max}(\mathbb{E}[gg^\mathsf{T}])$ for all $i,j$ and apply Lemma 1. To see this, note that $D_{i,j}$ can be represented by a Rayleigh quotient, i.e.,

$$
D_{i,j} = \mathbb{E}[G_{i,j}^2] = e_{i,j}^\mathsf{T}\mathbb{E}[gg^\mathsf{T}]e_{i,j} \tag{Adam}
$$
$$
D_{i,j} = \mathbb{E}[(Q_L^\mathsf{T} G Q_R)_{i,j}^2] = e_{i,j}^\mathsf{T}(Q_R \otimes Q_L)^\mathsf{T}\mathbb{E}[gg^T](Q_R \otimes Q_L)e_{i,j} \tag{EShampoo}
$$

where $e_{i,j} = \text{vec}\, E_{i,j} \in \mathbb{R}^{mn}$ and $E_{i,j} \in \mathbb{R}^{m \times n}$ with

$$
(E_{i,j})_{k,l} = \begin{cases} 1 & \text{if } (k,l) = (i,j) \\ 0 & \text{otherwise.} \end{cases}
$$

Since $\|e_{i,j}\|_2 = \|(Q_R \otimes Q_L)e_{i,j}\|_2 = 1$, the desired bound by the extreme eigenvalues follows from the Courant–Fischer–Weyl min-max theorem.

To prove Equation (22), observe that under independence between components of the gradient $G_{i,j}$, the preconditioner for full-matrix Adam $\mathbb{E}[gg^\mathsf{T}]$ is a diagonal matrix whose diagonal entries consist of $\mathbb{E}[G_{i,j}^2]$ for all $i,j$. Hence, $\mathbb{E}[G_{i,j}^2]$ is bounded by the minimum and maximum eigenvalues, i.e., $\mathbb{E}[G_{i,j}^2] \in [\lambda_{\min}(\mathbb{E}[gg^\mathsf{T}]), \lambda_{\max}(\mathbb{E}[gg^\mathsf{T}])]$.

Due to independence, $\mathbb{E}[GG^\mathsf{T}]$ and $\mathbb{E}[G^\mathsf{T} G]$ are also diagonal. Expanding their diagonal entries gives

$$
(\mathbb{E}[GG^\mathsf{T}])_{i,i} = \sum_{j=1}^{n} \mathbb{E}[G_{i,j}^2] \in n \cdot [\lambda_{\min}(\mathbb{E}[gg^\mathsf{T}]), \lambda_{\max}(\mathbb{E}[gg^\mathsf{T}])] \qquad \forall\, i = 1, ..., m,
$$

$$
(\mathbb{E}[G^\mathsf{T} G])_{j,j} = \sum_{i=1}^{m} \mathbb{E}[G_{i,j}^2] \in m \cdot [\lambda_{\min}(\mathbb{E}[gg^\mathsf{T}]), \lambda_{\max}(\mathbb{E}[gg^\mathsf{T}])] \qquad \forall\, j = 1, ..., n.
$$

Therefore, the Shampoo preconditioner $(\mathbb{E}[GG^\mathsf{T}] \otimes \mathbb{E}[G^\mathsf{T} G])^{1/2}$ is also diagonal with eigenvalues lying in the interval $[m^{1/2}n^{1/2}\lambda_{\min}(\mathbb{E}[gg^\mathsf{T}]), m^{1/2}n^{1/2}\lambda_{\max}(\mathbb{E}[gg^\mathsf{T}])]$. The desired result follows.

$\square$

Note that more general bounds can be derived by relaxing the assumption $\mathbb{E}[G] = 0$, although the bounds are more complex and are not more conceptually informative.

# D On the gap between the optimal and practical eigenvalue correction

Eigenvalue-corrected Shampoo shares similarities with memory-efficient optimizers such as GaLore, which apply Adam in a low-dimensional subspace spanned by the largest singular vectors of the gradient matrix (Zhao et al., 2024; Su et al., 2025). However, GaLore relies on the assumption that the gradient resides in a low-rank subspace that evolves gradually, allowing the same optimizer state to be used even as the subspace is updated. A recent method called LDAdam proposed by Robert et al. (2024) describes a projection-aware method that corrects the scaling matrix through both a projection-aware update and generalized error feedback mechanism when transitioning between subspaces to address this issue. We will see that the naive eigenvalue correction as used in SOAP suffers from similar limitations.

## D.1 Optimal eigenvalue correction in Frobenius norm

Note that we can determine the optimal eigenvalue correction by minimizing its Frobenius norm approximation to full-matrix AdaGrad or Adam:

**Proposition 2.** *Given symmetric matrix $C \in \mathbb{R}^{d \times d}$ and orthogonal matrix $Q \in \mathbb{R}^{d \times d}$, the optimal eigenvalue correction $D^*$ that minimizes the Frobenius norm distance is given by:*

$$D^* := \mathrm{diag}(Q^\mathsf{T} C Q) \in \arg\min_{D \in \mathbb{D}^d} ||C - QDQ^\mathsf{T}||_{\mathrm{F}}.^{[11]} \tag{23}$$

*The exact expression for $D^*$ depends on the form of accumulation used for computing $C$; note that the damping term is omitted here since it does not change the optimal solution. $T$ denotes the current iteration.*

1. *(Idealized) If $C = \mathbb{E}[gg^\mathsf{T}]$, then $D^* = \mathbb{E}[\mathrm{diag}(Q^\mathsf{T}g)^{\odot 2}]$ (George et al., 2018).*

2. *(AdaGrad) If $C = \sum_{t=1}^{T} g_t g_t^\mathsf{T}$, then $D_T^* = \sum_{t=1}^{T} \mathrm{diag}(Q_T^\mathsf{T} g_t)^{\odot 2}$.*

3. *(Adam) If $C = (1 - \beta_2) \sum_{t=1}^{T} \beta_2^{T-t} g_t g_t^\mathsf{T}$, then $D_T^* = (1 - \beta_2) \sum_{t=1}^{T} \beta_2^{T-t} \mathrm{diag}(Q_T^\mathsf{T} g_t)^{\odot 2}$. Note that $C$ can be generated recursively via an exponential moving average $C_T = \beta_2 C_{T-1} + (1 - \beta_2) g_T g_T^\mathsf{T}$ with $C_0 = 0 \in \mathbb{R}^{mn \times mn}$.*

*Proof.* (Informal.) Since $Q$ is orthogonal, $||C - QDQ^\mathsf{T}||_F = ||Q^\mathsf{T} C Q - D||_F$, with $D$ diagonal. Therefore, the optimal solution has the form $D^* = \mathrm{diag}(Q^\mathsf{T} C Q)$. Each case then follows by observing that $C$ is the expectation or weighted sum of $gg^\mathsf{T}$, and passing $Q$ into the sum or expectation. □

While $D^*$ is optimal in this sense, it does not guarantee anything regarding the similarity of the (root) inverse of the approximation. Using Proposition 2, we can establish the following corollary.

**Corollary 1.** *The optimal eigenvalue correction yields a tighter Frobenius norm approximation than Shampoo and CASPR, i.e.,*

$$||C - QD^*Q^\mathsf{T}||_{\mathrm{F}} \leq \min\{||C - C^{\mathrm{Shampoo}}||_{\mathrm{F}}, ||C - C^{\mathrm{CASPR}}||_{\mathrm{F}}\},$$

*where $C^{\mathrm{CASPR}}$ is the preconditioner used by the CASPR algorithm (Duvvuri et al., 2024).*

*Proof.* (Informal.) The first inequality trivially follows from Proposition 2 since by definition the eigenvectors of Shampoo are equal to $Q = (Q_R \otimes Q_L)^\mathsf{T}$. The second inequality follows from Proposition 2 together with Lemma 3.4 in Duvvuri et al. (2024), which states that the eigenvectors of $C^{\mathrm{Shampoo}}$ and $C^{\mathrm{CASPR}}$ are identical. □

This means that using the optimal eigenvalue correction (with respect to Frobenius norm approximation) yields a tighter approximation to the full-matrix quantity compared to Shampoo or CASPR. A remaining open theoretical question is whether a tighter Frobenius norm approximation can yield a tighter regret bound compared to Shampoo and CASPR. For example, one can obtain a tighter regret bound than Shampoo by only considering one-sided Shampoo, which is not generally a better approximation to full-matrix AdaGrad (Xie et al., 2025; An et al., 2025).

---

[11]$\mathbb{D}^d$ denotes the set of $d \times d$ diagonal matrices.

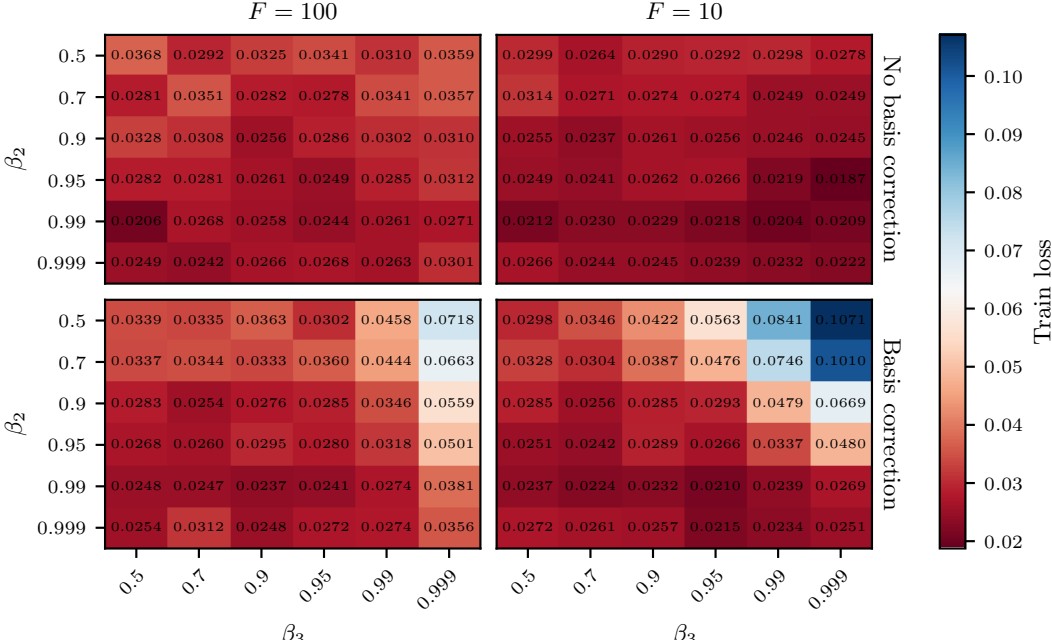

Figure 6: Decoupling the exponential moving average for the eigenbasis ($\beta_2$) and eigenvalues ($\beta_3$) for different eigendecomposition frequencies ($F$). We can observe a remarkable invariance to the choice of $\beta_2$ and $\beta_3$. Interestingly, the correction for the change of bases seems to *hurt* performance overall and especially for low $\beta_2$, high $\beta_3$, and low $F$ values – a pattern that we might expect *without* the correction.

### D.2    Basis-aware eigenvalue correction

Despite its optimality, computing $D^*$ for cases 2 and 3 in Proposition 2 is not feasible in practice since we would have to store and transform prior gradients from *all previous iterations* with $Q_T$, or have access to the full-matrix quantity $C_T$. For simplicity, we will focus on case 3, although the results generalize without loss of generality.[12] In contrast to the optimal eigenvalue correction, which uses a fixed basis matrix $Q_T$ based on the most recent statistics $L_T$ and $R_T$, SOAP uses potentially different basis matrices $Q_t$ based on previous statistics $L_t$ and $R_t$ available at every step $t = 1, ..., T$:

$$D^* = (1 - \beta_2) \sum_{t=1}^{T} \beta_2^{T-t} \operatorname{diag}\left(Q_T^\mathsf{T} g_t\right)^{\odot 2} \approx (1 - \beta_2) \sum_{t=1}^{T} \beta_2^{T-t} \operatorname{diag}\left(Q_t^\mathsf{T} g_t\right)^{\odot 2} =: \hat{D}_T. \quad (24)$$

Intuitively, this means that the eigenvalue correction is accumulated inconsistently *across different coordinate systems*. This can lead to a mismatch when preconditioning $G_T$, which is only transformed to the *current* coordinate system determined by $Q_T$. When the basis remains approximately constant, i.e., $Q_T \approx Q_t$ for all $t$, Equation (24) can be a tight approximation and $D^* \approx D_T$.

The naive eigenvalue correction may be approximately correct when the basis is updated infrequently, but this can be a poor approximation if $Q$ does not approximate the changing eigenbasis of $C$, thereby increasing the approximation error. For case 3, the approximation becomes potentially milder because the terms in the sum in Equation (24) are down-weighted by $\beta_2^{T-t}$ through the exponential moving average. Therefore, depending on $\beta_2$, the contribution from the eigenvalue correction statistic in previous coordinate systems might be negligible. However, this is not the case for case 2, where all terms are weighted equally.

---

[12]To address case 2, simply drop $1 - \beta_2$ and $\beta_2^{T-t}$.

In order to address the theory-practice gap between the naive eigenvalue correction (used in SOAP and EShampoo) and the optimal correction in Frobenius norm, we first observe that

$$
\begin{aligned}
\boldsymbol{D}^* &= (1 - \beta_2) \sum_{t=1}^{T} \beta_2^{T-t} \, \mathrm{diag} \left( \boldsymbol{Q}^\intercal \boldsymbol{g}_t \right)^{\odot 2} \\
&= (1 - \beta_2) \sum_{t=1}^{T} \beta_2^{T-t} \, \mathrm{diag} \left( \boldsymbol{Q}_T^\intercal \boldsymbol{g}_t \boldsymbol{g}_t^\intercal \boldsymbol{Q}_T \right) \\
&= \mathrm{diag} \left( \boldsymbol{Q}_T^\intercal \big( (1 - \beta_2) \sum_{t=1}^{T} \beta_2^{T-t} \boldsymbol{g}_t \boldsymbol{g}_t^\intercal \big) \boldsymbol{Q}_T \right) \\
&= \mathrm{diag} \left( \boldsymbol{Q}_T^\intercal \boldsymbol{C}_T \boldsymbol{Q}_T \right) \\
&=: \mathrm{diag} \left( \hat{\boldsymbol{C}}_T \right).
\end{aligned}
\tag{25}
$$

Since $\boldsymbol{Q}_T$ is orthogonal, we can recursively write

$$
\begin{aligned}
\boldsymbol{D}^* = \mathrm{diag} \left( \hat{\boldsymbol{C}}_T \right) &= \mathrm{diag} \left( \boldsymbol{Q}_T^\intercal (\beta_2 \boldsymbol{C}_{T-1} + (1 - \beta_2) \boldsymbol{g}_T \boldsymbol{g}_T^\intercal) \boldsymbol{Q}_T \right) \\
&= \mathrm{diag} \left( \beta_2 \boldsymbol{Q}_T^\intercal \boldsymbol{C}_{T-1} \boldsymbol{Q}_T + (1 - \beta_2) \boldsymbol{Q}_T^\intercal \boldsymbol{g}_t \boldsymbol{g}_t^\intercal \boldsymbol{Q}_T \right) \\
&= \mathrm{diag} \left( \beta_2 \boldsymbol{Q}_T^\intercal \boldsymbol{Q}_{T-1} \hat{\boldsymbol{C}}_{T-1} \boldsymbol{Q}_{T-1}^\intercal \boldsymbol{Q}_T + (1 - \beta_2) \boldsymbol{Q}_T^\intercal \boldsymbol{g}_T \boldsymbol{g}_T^\intercal \boldsymbol{Q}_T \right) \\
&= \beta_2 \, \mathrm{diag} \left( \boldsymbol{R}_{T,T-1} \hat{\boldsymbol{C}}_{T-1} \boldsymbol{R}_{T,T-1}^\intercal \right) + (1 - \beta_2) \, \mathrm{diag} \left( \boldsymbol{Q}_T^\intercal \boldsymbol{g}_T \right)^{\odot 2},
\end{aligned}
\tag{26}
$$

where $\boldsymbol{R}_{T,T-1} := \boldsymbol{Q}_T^\intercal \boldsymbol{Q}_{T-1}$ is the transition matrix between different bases.

While this is the exact expression for our desired quantity, it requires keeping track of the full matrix $\hat{\boldsymbol{C}}_{T-1}$ to compute $\mathrm{diag}(\boldsymbol{R}_{T,T-1} \hat{\boldsymbol{C}}_{T-1} \boldsymbol{R}_{T,T-1}^\intercal)$, which is not tractable. Since we explicitly construct $\boldsymbol{Q}_{T-1}$ to be close to the best Kronecker-factored basis for $\boldsymbol{C}_{T-1}$ through the choice of the Shampoo preconditioner with $\boldsymbol{L}_{T-1}$ and $\boldsymbol{R}_{T-1}$, we make the additional assumption that $\hat{\boldsymbol{C}}_{T-1}$ is approximately diagonal, i.e., $\hat{\boldsymbol{C}}_{T-1} \approx \mathrm{diag}(\hat{\boldsymbol{C}}_{T-1}) = \mathrm{diag}(\boldsymbol{v}_{T-1}^{\mathrm{corrected}})$.

Substituting this back into the recursive equation above, we have that

$$
\begin{aligned}
\boldsymbol{D}^* = \mathrm{diag} \left( \hat{\boldsymbol{C}}_T \right) &\approx \beta_2 \, \mathrm{diag} \left( \boldsymbol{R}_{T,T-1} \, \mathrm{diag} \left( \boldsymbol{v}_{T-1}^{\mathrm{corrected}} \right) \boldsymbol{R}_{T,T-1}^\intercal \right) + (1 - \beta_2) \, \mathrm{diag} \left( \boldsymbol{Q}_T^\intercal \boldsymbol{g}_T \right)^{\odot 2} \\
&= \beta_2 \, \mathrm{diag} \left( \boldsymbol{R}_{T,T-1}^{\odot 2} \boldsymbol{v}_{T-1}^{\mathrm{corrected}} \right) + (1 - \beta_2) \, \mathrm{diag} \left( \boldsymbol{Q}_T^\intercal \boldsymbol{g}_T \right)^{\odot 2} \\
&= \mathrm{diag} \left( \boldsymbol{v}_T^{\mathrm{corrected}} \right).
\end{aligned}
\tag{27}
$$

This gives a recursive update rule that, in contrast to the naive solution, accounts for the changes of bases between iterations. This also recovers the exact solution when $\boldsymbol{Q}_T = \boldsymbol{Q}_t$ for all $t$ since $\boldsymbol{R}_{t,t-1} = \mathbf{I}$. This correction for the changing basis has a similar motivation to and resembles parts of the LDAdam algorithm (Robert et al., 2024).

We design an experiment to empirically test whether the implicit approximation in Equation (24) helps in practice, and find that interestingly, the correction appears to hurt performance in the settings where we would expect it to help, see Figure 6. A satisfying explanation of this phenomenon remains an open question.

## E   Additional experimental details and results

For all experiments we used $1\times$ NVIDIA A100 80GB GPU per run, with the exception of the ImageNet ViT experiments, for which we used $4\times$ NVIDIA A100 80GB GPUs per run. All of the experiments were conducted on an internal compute cluster and we estimate that the total required compute was around 1440 GPU hours or 60 days. We ran additional exploratory experiments not reported in this paper which increases the total compute cost of the project. The implementation of EShampoo and all other Shampoo variants considered here including Algorithm 4 and Equation (11) for `eigh` is available at https://github.com/facebookresearch/optimizers.

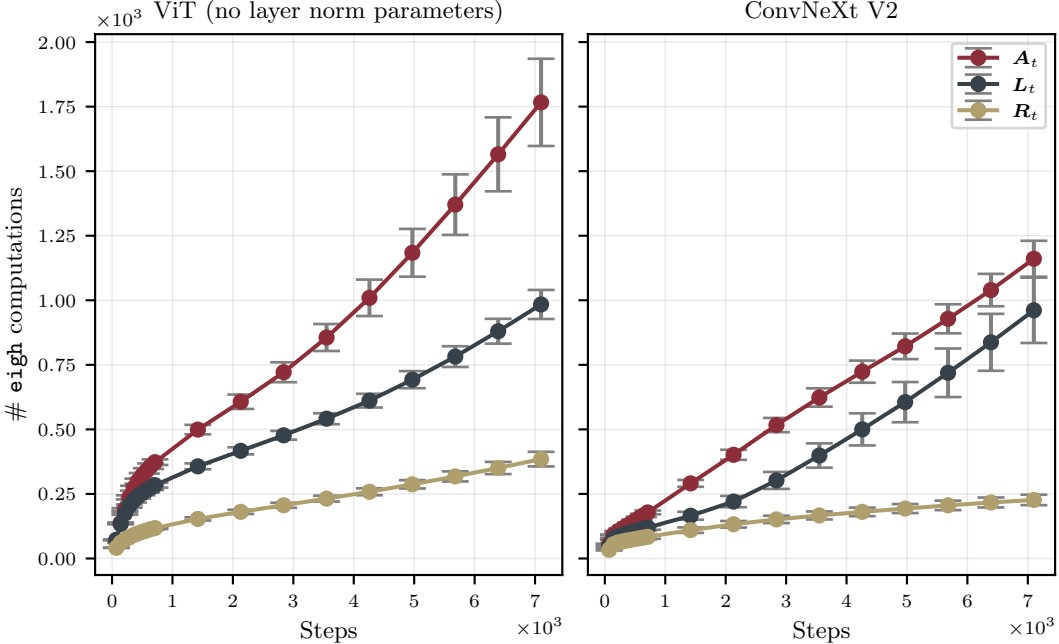

Figure 7: Same setting as in Figure 4. (**left**) When removing all learnable layer norm parameters from the ViT, the number of eigendecompositions for $L_t$ and $R_t$ remain approximately unchanged and no compensation appears to be happening. (**right**) With a ConvNeXt V2 architecture instead of the ViT, the overall pattern is similar, but with a more pronounced discrepancy between $L_t$ and $R_t$.

## E.1 Imagewoof experimental details

We use the Imagewoof dataset.[13] All models are trained with cross entropy loss for 100 epochs, using a learning rate schedule consisting of a linear warmup for 353 steps followed by cosine decay. We use a batch size of 128, randomized cropping and horizontal flips as data augmentation, and the default settings for $\beta_1 = 0.9$ and $\beta_2 = 0.999$. For EShampoo in Figure 3, Figure 4, and Figure 5 we use the learning rate $\alpha = 6 \cdot 10^{-4}$ and $\epsilon = 10^{-10}$.

For the vision transformer (ViT) model, we use SimpleViT (Beyer et al., 2022) with patch size 16, 6 heads, a depth of 12 layers, an MLP dimension of 1536, dimension of 384, gradient clipping with threshold 1, and weight decay of $10^{-4}$. For the ConvNeXt V2 architecture (Woo et al., 2023) we use weight decay of 0.05 and drop paths with rate 0.1.

For Figure 2, we fix $\epsilon = 10^{-10}$ and sweep the following learning rates $\alpha$:[14]

- Vision transformer
    - AdamW: $\alpha \in \{10^{-4}, 3 \cdot 10^{-4}, 6 \cdot 10^{-4}, 10^{-3}, 3 \cdot 10^{-3}, 6 \cdot 10^{-3}, 10^{-2}\}$
    - Shampoo with grafting: $\alpha \in \{10^{-4}, 3 \cdot 10^{-4}, 10^{-3}, 3 \cdot 10^{-3}\}$
    - Shampoo: $\alpha \in \{3 \cdot 10^{-3}, 6 \cdot 10^{-3}, 10^{-2}, 3 \cdot 10^{-3}\}$
    - Shampoo$^2$ with trace scaling: $\alpha \in \{10^{-4}, 3 \cdot 10^{-4}, 6 \cdot 10^{-4}, 3 \cdot 10^{-3}\}$
    - EShampoo: $\alpha \in \{10^{-4}, 3 \cdot 10^{-4}, 10^{-3}, 3 \cdot 10^{-3}\}$
- ConvNeXt V2
    - AdamW: $\alpha \in \{10^{-4}, 3 \cdot 10^{-4}, 6 \cdot 10^{-4}, 10^{-3}, 3 \cdot 10^{-3}, 6 \cdot 10^{-3}, 10^{-2}\}$
    - Shampoo with grafting: $\alpha \in \{10^{-4}, 3 \cdot 10^{-4}, 10^{-3}, 3 \cdot 10^{-3}, 10^{-2}\}$
    - Shampoo: $\alpha \in \{10^{-4}, 3 \cdot 10^{-4}, 10^{-3}, 3 \cdot 10^{-3}\}$
    - Shampoo$^2$ with trace scaling: $\alpha \in \{10^{-4}, 3 \cdot 10^{-4}, 10^{-3}\}$
    - EShampoo: $\alpha \in \{10^{-4}, 3 \cdot 10^{-4}, 10^{-3}, 3 \cdot 10^{-3}, 10^{-2}\}$

---

[13]The Imagewoof dataset is available at https://github.com/fastai/imagenette.

[14]We fixed $\epsilon$ based on a wide sweep over $\alpha$ and $\epsilon$ for AdamW. We also use this $\epsilon$ when grafting from Adam.

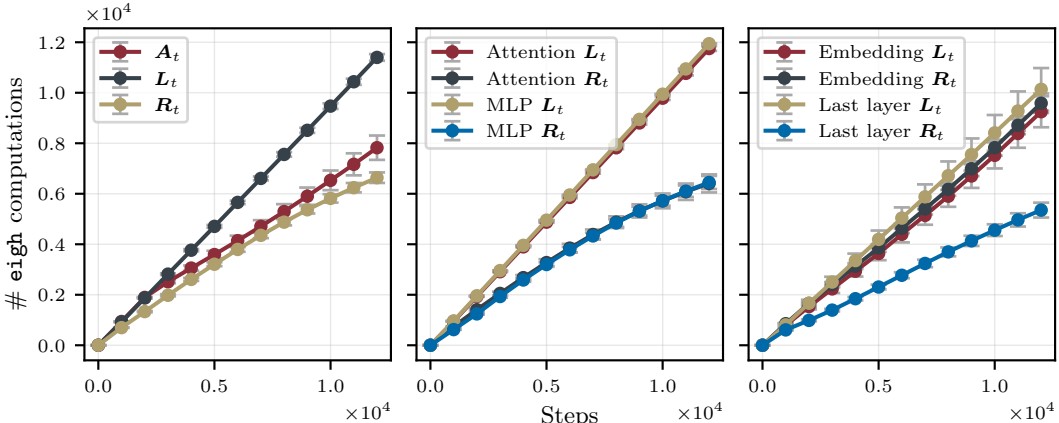

Figure 8: Llama 3 (324M) trained on 3.2B tokens of C4 data with EShampoo ($F = 1, \tau = 0.01$). The eigenbases for $\boldsymbol{L}_t$ are also in this setting consistently updated more frequently than for $\boldsymbol{R}_t$, with the exception of the embedding. The eigenbases of $\boldsymbol{A}_t$ are less frequently updated than of $\boldsymbol{L}_t$, in contrast to the Imagewoof experiments. There appears to be no difference in the pattern for weight matrices within the attention mechanisms and the MLPs.

### E.2   More patterns of adaptivity

To test the generality of the trends described in Section 4.3, we perform the same experiment under a few additional settings.

In Figure 7, we remove all learnable layer norm parameters from the vision transformer and consider the ConvNeXt V2 model trained on the same dataset (Imagewoof). The overall pattern is consistent across all architectures. When removing the learnable layer norm parameters from the vision transformer, the number of eigendecompositions for $\boldsymbol{L}_t$ and $\boldsymbol{R}_t$ stays roughly constant (compare Figure 4 (right) with Figure 7 (left)). For the ConvNeXt V2 architecture, the discrepancy between the required updates for $\boldsymbol{L}_t$ and $\boldsymbol{R}_t$ is even more pronounced: on average, the eigenbases of $\boldsymbol{L}_t$ has to be updated significantly more frequently than for $\boldsymbol{R}_t$.

To expand beyond the vision modality and cover class imbalance, we also train a Llama 3 model with 324 million parameters on 3.2 billion tokens of the C4 dataset and conduct a similar analysis (Raffel et al., 2023; Grattafiori et al., 2024). We use EShampoo with $F = 1$ and $\tau = 0.01$.

In Figure 8 (left), we compute mean and standard errors across all (single) Kronecker factors $\boldsymbol{A}_t$ for RMS normalization parameters and all Kronecker factors $\boldsymbol{L}_t$ and $\boldsymbol{R}_t$ for linear layers and embedding blocks at every iteration. There are no bias parameters in this particular model. Consistent with the experiments in the vision setting, $\boldsymbol{L}_t$ is updated more frequently than $\boldsymbol{R}_t$, with $\boldsymbol{L}_t$ updated at almost every iteration, whereas $\boldsymbol{R}_t$ updated every other iteration. Unlike the vision setting, the Kronecker factors for the normalization layers $\boldsymbol{A}_t$ are initially updated at a similar frequency to $\boldsymbol{L}_t$, but diminishes after the first 25% of iterations until it is closer to, but still higher than, the update frequency for $\boldsymbol{R}_t$. The standard error for $\boldsymbol{A}_t$ is also larger than for $\boldsymbol{L}_t$ and $\boldsymbol{R}_t$.

In Figure 8 (middle), we consider two subsets of the hidden layers: the four weight matrices in the attention mechanism in each transformer block and the three weight matrices in the MLP in each transformer block. We compute the statistics across all weight matrices for each subset of the hidden weight matrices. There appears to be no significant difference in the number of eigendecompositions across steps between the two subsets of the weights.

In Figure 8 (right), we consider the input embedding layer and the last output layer. Due to the large vocabulary size, the gradients for these two layers are blocked such that no block has a dimension larger than 8192. Then we precondition each gradient block as usual with $\boldsymbol{L}_t$ and $\boldsymbol{R}_t$. Here, we compute the statistics across all of these blocks. The trend for the last layer is consistent with all other linear layers in our experiments. However, the eigendecomposition frequency of $\boldsymbol{L}_t$ and $\boldsymbol{R}_t$ for embeddings are almost identical.

Table 3: Results for all considered settings on a subset of the AlgoPerf workloads. We show the mean and standard error of the steps and time to the targets across the runs that actually hit the targets.

| Workload | Shampoo Variant | Hits Targets | Steps | Time [min] |
|---|---|---|---|---|
| FastMRI | Adam grafting ($F = 100$) | 4/5 | $4301 \pm 109$ | $13.96 \pm 0.44$ |
| | $C^{\mathrm{EShampoo}}$ ($F = 100$) | 5/5 | $2536 \pm 66$ | $10.44 \pm 0.21$ |
| | $C^{\mathrm{EShampoo}}$ ($F = 50$) | 5/5 | $2578 \pm 86$ | $10.86 \pm 0.27$ |
| | $C^{\mathrm{EShampoo}}$ ($F = 10$) | 5/5 | $2311 \pm 73$ | $14.93 \pm 1.97$ |
| | $C^{\mathrm{EShampoo}}$ ($F = 1$) | 5/5 | $2101 \pm 31$ | $35.34 \pm 0.70$ |
| | $C^{\mathrm{EShampoo}}$ ($\tau = 0.1, F = 100$) | 5/5 | $2553 \pm 154$ | $16.33 \pm 2.10$[17] |
| | $C^{\mathrm{EShampoo}}$ ($\tau = 0.1, F = 50$) | 5/5 | $2468 \pm 145$ | $10.81 \pm 0.72$ |
| | $C^{\mathrm{EShampoo}}$ ($\tau = 0.1, F = 10$) | 5/5 | $2420 \pm 92$ | $10.76 \pm 0.73$ |
| | $C^{\mathrm{EShampoo}}$ ($\tau = 0.1, F = 1$) | 5/5 | $2367 \pm 96$ | $10.93 \pm 0.42$ |
| | $C^{\mathrm{EShampoo}}$ ($\tau = 0.01, F = 1$) | 5/5 | $2208 \pm 41$ | $27.43 \pm 0.49$ |
| ImageNet ViT | Adam grafting ($F = 100$) | 1/1 | 79907 | 894.27 |
| | $C^{\mathrm{EShampoo}}$ ($F = 100$) | 1/1 | 76226 | 894.85 |
| | $C^{\mathrm{EShampoo}}$ ($F = 10$) | 1/1 | 73237 | 1160.53 |
| | $C^{\mathrm{EShampoo}}$ ($\tau = 0.1, F = 100$) | 1/1 | 74010 | 852.66 |
| | $C^{\mathrm{EShampoo}}$ ($\tau = 0.1, F = 50$) | 1/1 | 77459 | 935.89 |
| | $C^{\mathrm{EShampoo}}$ ($\tau = 0.01, F = 10$) | 1/1 | 75841 | 1188.76 |
| OGBG | Adam grafting ($F = 100$) | 2/5 | $12574 \pm 708$ | $39.20 \pm 1.88$ |
| | $C^{\mathrm{EShampoo}}$ ($F = 100$) | 3/5 | $8320 \pm 1203$ | $33.02 \pm 4.05$ |
| | $C^{\mathrm{EShampoo}}$ ($F = 50$) | 5/5 | $7173 \pm 443$ | $26.17 \pm 1.31$ |
| | $C^{\mathrm{EShampoo}}$ ($F = 10$) | 3/5 | $6645 \pm 357$ | $37.55 \pm 1.74$ |
| | $C^{\mathrm{EShampoo}}$ ($F = 1$)[18] | — | — | — |
| | $C^{\mathrm{EShampoo}}$ ($\tau = 0.1, F = 100$) | 4/5 | $8047 \pm 369$ | $27.60 \pm 1.15$ |
| | $C^{\mathrm{EShampoo}}$ ($\tau = 0.1, F = 50$) | 5/5 | $7117 \pm 328$ | $27.55 \pm 3.49$ |
| | $C^{\mathrm{EShampoo}}$ ($\tau = 0.1, F = 10$) | 5/5 | $7151 \pm 416$ | $29.11 \pm 1.98$ |
| | $C^{\mathrm{EShampoo}}$ ($\tau = 0.1, F = 1$) | 5/5 | $6758 \pm 273$ | $34.16 \pm 1.65$ |
| | $C^{\mathrm{EShampoo}}$ ($\tau = 0.01, F = 10$) | 2/5 | $7234 \pm 361$ | $39.15 \pm 2.01$ |

### E.3 AlgoPerf workloads

We follow the standard AlgoPerf setup and consider wall-clock time to pre-specified validation metric targets. See Dahl et al. (2023) and Kasimbeg et al. (2025) for more details on the AlgoPerf benchmark.[15] The FastMRI dataset can be attributed to Knoll et al. (2020); Zbontar et al. (2019), the ImageNet dataset to Krizhevsky et al. (2012), and the OGBG dataset to Hu et al. (2021).

We choose this specific subset of the workloads because (1) we want to include a larger scale vision transformer (ImageNet ViT), an architecture we use in the small-scale experiments, (2) the FastMRI and OGBG workloads share the same hyperparameter settings with the Imagenet ViT workload, hence excluding them as a confounding factor, and (3) the $\beta_2$ used for these workloads is the smallest among all hyperparameter settings of the winning Shampoo submission, resulting in the fastest moving average and thereby potentially faster changing eigenbases.

We run the winning Shampoo submission with Adam grafting, $F = 100$, and the best hyperparameter setting for each workload.[16] For EShampoo, we use the same best-performing hyperparameter setting from Shampoo but turn off learning rate grafting from Adam, and modify $F$ and $\tau$.

---

[15]The benchmark code is available at `https://github.com/mlcommons/algorithmic-efficiency/`.

[16]The submission is available at `https://github.com/mlcommons/submissions_algorithms/tree/main/previous_leaderboards/algoperf_v05/submissions/external_tuning/shampoo_submission`.

[17]This wall-clock time statistic seems to be negatively impacted by either an issue with the AlgoPerf code or the hardware setup.

[18]Runs failed due to `https://github.com/mlcommons/algorithmic-efficiency/issues/866`.

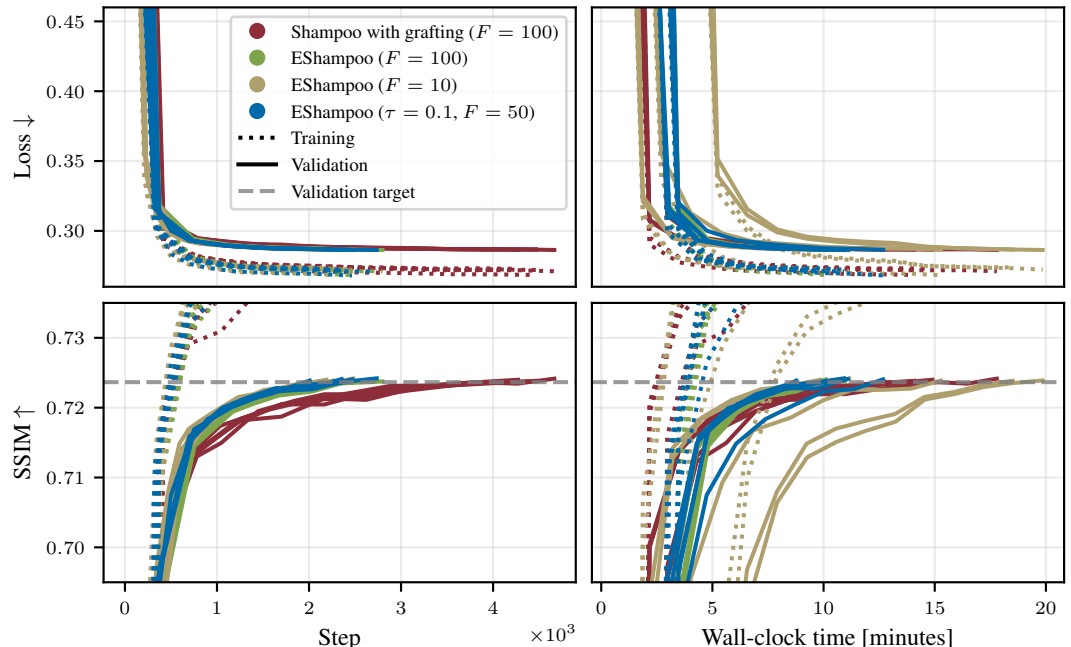

Figure 9: FastMRI AlgoPerf workload.

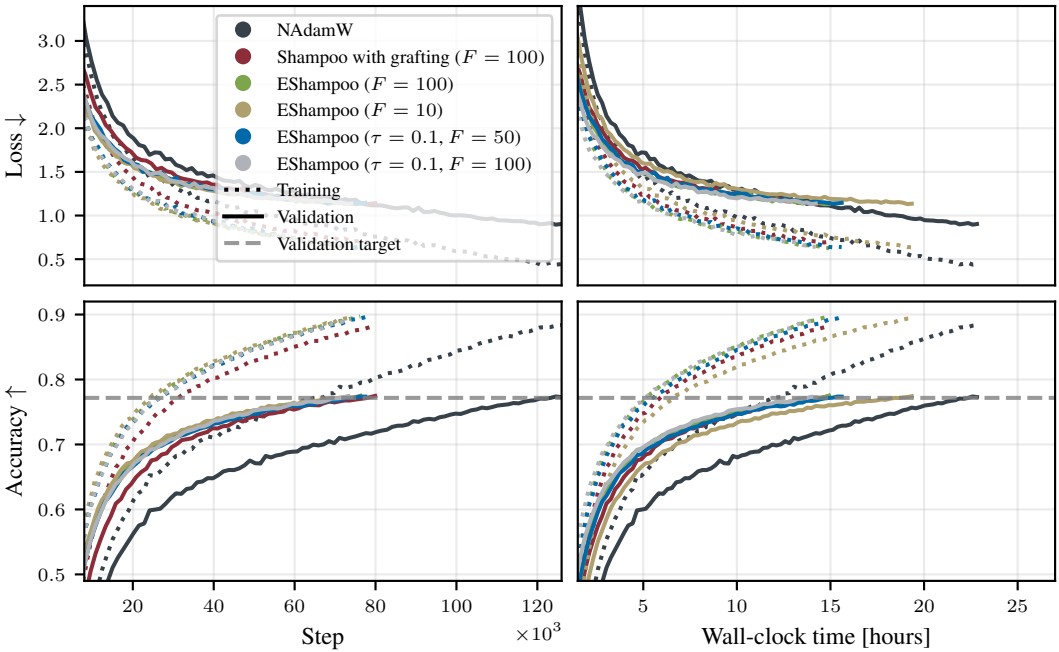

Figure 10: ImageNet ViT AlgoPerf workload.

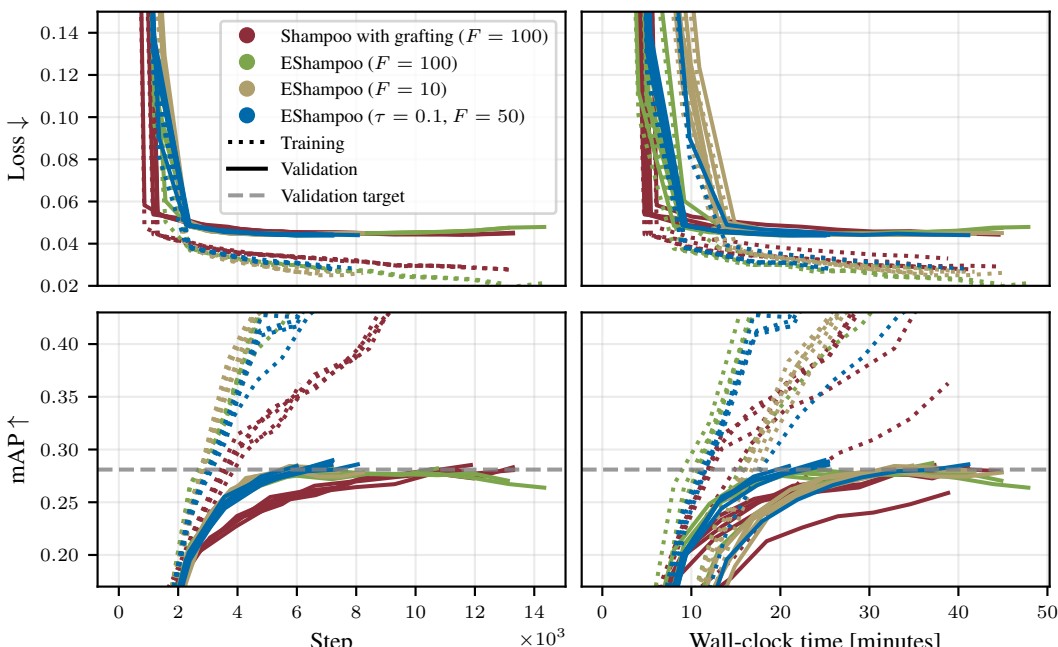

Figure 11: OGBG AlgoPerf workload.

