# OpenReview forum: "Purifying Shampoo: Investigating Shampoo's Heuristics by Decomposing its Preconditioner"
_NeurIPS.cc/2025/Conference — NeurIPS 2025 spotlight_

### Official Review · Reviewer_2S8j · 2025-06-28

**Clarity:** 3
**Significance:** 3
**Originality:** 4
**Rating:** 5
**Confidence:** 4

**Summary:**

This paper studies the practical tricks of the optimizer Shampoo to address the critical problem of stale updates. The paper demonstrates theoretically that the stale update of Shampoo can lead to eigenevalue scale mismatch, resulting in significantly worse performance, which can be fixed by tricks like grafting and correcting eigenvalues (e.g., SOAP). Then the authors discuss the effect of stale eigenbasis and propose an adaptive way to update the eigenbasis to solve the problem. Empirical results are also provided to verify the claims.

**Questions:**

1. Based on the explanation for why grafting is useful for Shampoo, I am curious why don't we just use $\sqrt{mn}$ instead of $||U_t^{Adam}||_F$ in grafting for a matrix $W\in\mathbb{R}^{m\times n}$, since they should have similar scales. Have you tried this kind of trivial scaling?
2. I want to double check what the wall-clock time stands for in e.g., Figure 3. Is it the total training time including forward and backward or any other measure? Also, could you post the proportion of the optimizer step time in the whole training process? Based on the figure, the optimizer step computation takes the major time costs, which seems a bit strange, even though matrix operations are required.
3. The observation that $L_t$ seems to update more than $R_t$ looks interesting. Do you have more evidence for this in other models or other scales? Also, any intuition for explaining this would be welcome.

**Ethical Concerns:**

["NO or VERY MINOR ethics concerns only"]

**Final Justification:**

Based on the rebuttal, my major concerns are addressed. I would raise my score to 5 since the paper explores some interesting and intuitive facts about the Shampoo optimizer, and the empirical results are solid.

**Limitations:**

Yes.

**Paper Formatting Concerns:**

No.

**Quality:**

4

**Strengths And Weaknesses:**

Strengths:
1. The paper is well-written and basically clear for understanding.
2. The viewpoint of eigenvalue scale mismatch for explaining the effectiveness of grafting and SOAP is interesting, providing a novel and valid perspective for the motivation of these Shampoo variants.
3. The study of the influence of the preconditioner staleness is detailed and solid, providing a comprehensive understanding of the problem.

Weakness:
1. The theoretical perspective in Section 3 on why grafting is useful may not be that rigorous and convincing. The bound in Lemma 1 and Proposition 1 is very loose, since the largest entry value/singular value can be much larger than the smallest ones. I don't think this bound is valid for justifying that the eigenvalue scale mismatch is the problem of Shampoo without grafting.
2. In Section 4, the adaptive update rule of the eigenbasis is claimed to propose for reducing the effort of manually tuning the update frequency. However, it turns out that the adaptive threshold $\tau$ also needs tuning and I don't see strong evidence in that the optimal choice can transfer over models and scales. This may make the idea not that powerful.

---

> ### Author Rebuttal · Authors · 2025-07-31
>
> Thank you for your comments and interesting questions. We address the weaknesses and questions below. We unfortunately cannot share figures for the new experimental results due to changes in rules this year. However, we will add all new results to the paper.
>
>
> > The theoretical perspective in Section 3 on why grafting is useful may not be that rigorous and convincing. The bound in Lemma 1 and Proposition 1 is very loose, since the largest entry value/singular value can be much larger than the smallest ones. I don't think this bound is valid for justifying that the eigenvalue scale mismatch is the problem of Shampoo without grafting.
>
> We agree that the bound is not tight. However, our aim is not to prove the tightest bound possible, but to demonstrate Shampoo's intrinsic dependence on the parameter dimensions $m, n$ to **motivate our experimental study** on Shampoo's layer-wise mis-scaling. While it is possible to derive tighter, more complex bounds with weaker assumptions, we felt that this would digress from the primary goal of this section.
>
>
> >In Section 4, the adaptive update rule of the eigenbasis is claimed to propose for reducing the effort of manually tuning the update frequency. However, it turns out that the adaptive threshold also needs tuning and I don't see strong evidence in that the optimal choice can transfer over models and scales. This may make the idea not that powerful.
>
> We do not claim that using the adaptive update rule reduces the need for manual hyperparameter tuning. However, tuning the threshold $\tau$ instead of an update frequency $F$ (both global scalar hyperparameters) enables us to directly control the approximation quality of each Kronecker factor and independently determine each factor's implicit update frequency schedule. Controlling the approximation quality directly lays the groundwork for studying its impact on convergence.
>
> From our perspective, two major challenges remain that prevent us from fully removing this hyperparameter (lines 309-312 in Section 5):
> 1. At this stage, we do not have a way of analyzing the impact of approximation quality or inexactness on Shampoo's convergence.
> 2. The wall-clock time overhead of the eigendecompositions relative to the rest of training is highly dependent on the training setup, e.g. model, training pipeline, and hardware.
>
>
> > Based on the explanation for why grafting is useful for Shampoo, I am curious why don't we just use $\sqrt{mn}$ instead of $||U_t^{Adam}||_F$ in grafting for a matrix $W \in \mathbb{R}^{m \times n}$, since they should have similar scales. Have you tried this kind of trivial scaling?
>
> Thank you for the suggestion! We have not tried this re-scaling previously. We expect that it can, depending on the setting, lead to stable training but Adam(W)'s optimal learning rate will generally not transfer.
>
> To test your idea, we **implemented this re-scaling** for the Imagewoof ViT workload. With AdamW's optimal learning rate the loss does not diverge, but the final solution is even worse than the one found with AdamW. However, by decreasing the learning rate by an order of magnitude, we can indeed reach close to the same final loss as with $||U_t^{Adam}||_F$ in this setting.
>
>
> > I want to double check what the wall-clock time stands for in e.g., Figure 3. Is it the total training time including forward and backward or any other measure? Also, could you post the proportion of the optimizer step time in the whole training process? Based on the figure, the optimizer step computation takes the major time costs, which seems a bit strange, even though matrix operations are required.
>
> The wall-clock time is total training time including data loading, forward and backward passes, and optimizer step. We have confirmed that the main overhead for this small model and dataset are indeed the eigendecompositions, which lead to drastic differences in the wall-clock time. Therefore, the exact proportion of the optimizer step in the overall training time strongly depends on the values of $F$ and $\tau$ (larger for small $F$/$\tau$).
>
>
> > The observation that $L_t$ seems to update more than $R_t$ looks interesting. Do you have more evidence for this in other models or other scales? Also, any intuition for explaining this would be welcome.
>
> Please refer to our response to Reviewer aEP3. We have performed a similar analysis to the one in Figure 4 for ConvNeXt V2 on the same dataset (Imagewoof) and a Llama 3 model with 160M parameters trained on 3.2B tokens of the C4 dataset, and found that the **pattern for $L_t$ and $R_t$ is consistent across all settings**.
>
> We do not have a strong explanation for this phenomenon at this time. One potentially helpful interpretation is viewing $L_t$ and $R_t$ as estimating the uncentered covariance of the gradients of the loss with respect to the layer's outputs and the uncentered covariance of the layer's inputs, respectively. (See Appendix B in [1] for more details.)
>
> [1] Anil et al. [Scalable Second Order Optimization for Deep Learning](https://arxiv.org/pdf/2002.09018) (2020)

---

> > ### Comment · Reviewer_2S8j · 2025-08-04
> > **Reply to the Author Rebuttal**
> >
> > Thanks for the reply and provided intuitions. My concerns are basically addressed. I would raise my score to 5 since the paper explores some interesting and intuitive facts about the Shampoo optimizer, and the empirical results are solid.

---

> > > ### Author Response · Authors · 2025-08-09
> > >
> > > We are glad that we could provide some further intuitions and thank you for raising your score.

---

### Official Review · Reviewer_QTer · 2025-06-30

**Clarity:** 3
**Significance:** 4
**Originality:** 2
**Rating:** 5
**Confidence:** 3

**Summary:**

The paper revisits practical tricks that have become standard when training with Shampoo: spacing out the eigendecomposition of its Kronecker factors and applying learning-rate grafting to match Adam’s step size. It shows that grafting compensates for eigenvalue issues: staleness between decompositions and a scale bias tied to layer shape, while the eigenvectors largely stay relevant. Replacing those stale eigenvalues each iteration with Adam-style second-moment estimates removes the need for grafting, and a simple Frobenius-norm test decides when the costlier eigenbasis must be refreshed. Experiments confirm that this “purified” variant matches or surpasses the grafted baseline with less compute.

**Questions:**

Could you please clarify how Eigenvalue-Corrected Shampoo meaningfully differs from SOAP? Appendix B of Anil et al. (2020) notes that the Frobenius-optimal diagonal Kronecker approximation in a fixed Shampoo eigenbasis is precisely the Adam second moment, and SOAP implements this by keeping the Shampoo basis (refreshed every F steps) while applying an Adam/Adafactor-style diagonal update at each iteration, so it seems EShampoo follows the exact same recipe.

**Ethical Concerns:**

["NO or VERY MINOR ethics concerns only"]

**Final Justification:**

The grafting analysis shows clear insight making the paper important and interesting.

**Limitations:**

Yes

**Quality:**

3

**Strengths And Weaknesses:**

The paper addresses a highly motivated problem. The paper offers an interesting explanation of why grafting is needed in standard Shampoo. By separating the Frobenius-norm effect of eigenvalues from the rotational effect of eigenvectors, they show that grafting merely rescales updates that have become too small or too large because the eigenvalues are both out-of-date. This clarifies a long-standing practical mystery.

While your analysis of grafting’s role in Shampoo and the supporting experiments are insightful, it’s still unclear what new algorithmic insight EShampoo contributes beyond SOAP, which already applies an Adam-style diagonal second moment in the fixed Shampoo eigenbasis.

---

> ### Author Rebuttal · Authors · 2025-07-31
>
> Thank you for the comments and clarifying questions. We will address the questions below.
>
>
> > While your analysis of grafting’s role in Shampoo and the supporting experiments are insightful, it’s still unclear what new algorithmic insight EShampoo contributes beyond SOAP, which already applies an Adam-style diagonal second moment in the fixed Shampoo eigenbasis.
>
> Our main objective in the first part of the paper (Section 3) is to demonstrate how eigenvalue correction methods (like SOAP) can be used to remove learning rate grafting which is a crucial heuristic in Shampoo -- we have not seen these observations discussed in-depth in the literature. We do not claim to propose eigenvalue-corrected Shampoo as a new algorithm and therefore **introduce it in the paper's background section** (Section 2.2); to our knowledge, eigenvalue correction was initially proposed in Liu et al. (2018) and George et al. (2018) for K-FAC and in Anil et al. (2020) for Shampoo.
>
>
> > Could you please clarify how Eigenvalue-Corrected Shampoo meaningfully differs from SOAP? Appendix B of Anil et al. (2020) notes that the Frobenius-optimal diagonal Kronecker approximation in a fixed Shampoo eigenbasis is precisely the Adam second moment, and SOAP implements this by keeping the Shampoo basis (refreshed every F steps) while applying an Adam/Adafactor-style diagonal update at each iteration, so it seems
>
> Thank you for the question. We unfortunately overloaded the term _eigenvalue-corrected Shampoo_ in the paper:
> 1. We use it to refer to the family of methods that apply an eigenvalue correction to Shampoo's preconditioner. Hence, SOAP is an instance of eigenvalue-corrected Shampoo.
> 2. We use its abbreviation EShampoo to refer to the method we implemented for the experiments. This is also an instance of eigenvalue-corrected Shampoo.
>
> We will clarify this terminology in our revision.
>
> From an algorithmic perspective, EShampoo as it is implemented in the experiments differs from SOAP in two respects (as discussed in Appendix B):
> 1. SOAP delays the update of the preconditioner until after the update of the weights.
> 2. SOAP uses a warm-started single iteration of the QR algorithm. EShampoo uses eigendecompositions via `torch.linalg.eigh` unless described otherwise (Figure 3 (left)).
>
> Regarding the optimality in Frobenius norm, as far as we know, **none of the practical implementations of eigenvalue corrections for Shampoo actually use the optimal solution** to approximate full-matrix Adam. SOAP/EShampoo (EShampoo referring to the method implemented for our experiments) make an additional approximation (Equation 20):
> $$(1 - \beta_2) \sum_{t=1}^T \beta_2^{T-t} \mathrm{diag}( Q_T^\mathrm{T} g_t )^{\odot 2} \approx (1 - \beta_2) \sum_{t=1}^T \beta_2^{T-t} \mathrm{diag}( Q_{t}^\mathrm{T} g_t )^{\odot 2}.$$
>
> Note the optimal solution (left) uses a constant $Q_T$ across iterations whereas the approximation (right) uses a different eigenbasis $Q_t$ for each iteration. This approach ignores the fact that we are accumulating gradients with respect to different eigenbases, especially when $F$ is small; see Appendix D.1 for further discussion of this approximation and an alternative eigenvalue correction.

---

### Official Review · Reviewer_aEP3 · 2025-07-01

**Clarity:** 4
**Significance:** 3
**Originality:** 3
**Rating:** 5
**Confidence:** 4

**Summary:**

This work studies the Shampoo optimizer and two key heuristics (learning rate grafting and preconditioning frequency) important for its competitive performance. Based on empirical results and theoretical motivation, the authors claim that the learning rate grafting addresses stale and mis-scaled eigenvalues of the preconditioner, and this need can be removed by decoupling the eigenvalue updates from the eigenbasis. The authors also propose a method tracking approximation-error of the Kronecker factors to adaptively determine eigenbasis update frequency. Their method also allows one to track which components of the network require more frequent eigendecomposition computations, showing that biases and layer normalization parameters require more frequent updates.

**Questions:**

* How would the authors’ proposed adaptive criterion of the QR algorithm compare to having frequency warmup (starting with frequency = 1 and roughly gradual increase to F = 50) on all of the parameters, or a fraction of the parameters (in terms of loss vs wall-clock time)?
* Do the authors have more intuition about the observed need for frequent updating for layer norm parameters? If the LayerNorms did not have learnable parameters, would the matrix layers then require more frequent computations (the LayerNorm parameters act as some output scale for the given layer)?

**Ethical Concerns:**

["NO or VERY MINOR ethics concerns only"]

**Final Justification:**

The authors have addressed my concerns regarding other domains, comparison to a frequency baseline warmup, and ablating the influence of LayerNorm parameters. I believe the empirical results are well-presented and analyses like these on Shampoo will be of interest to the community.

**Limitations:**

Yes

**Quality:**

3

**Strengths And Weaknesses:**

Strengths:
* The paper is clearly written and easy to follow. The relevant background and algorithms were presented particularly well.
* Connecting learning rate grafting to previously known modifications of Shampoo (using Shampoo^2 with trace scaling, low PF, eigenvalue correction a la SOAP) is a nice result.
* Investigations about determining eigenbasis computation frequency, its effect on different network parameters, and plotting tradeoffs between efficiency and performance is an important direction.

Weaknesses:
* The empirical setting is limited to two architectures on a single dataset and it would’ve been particularly interesting to see how the eigenbasis frequency results in eg. Figure 4 change for other domains, relating to previous work similarly identifying key network parameters requiring adaptivity due to issues like class imbalance for the last layer for instance [1,2].
* The authors’ proposed adaptive QR algorithm provides efficiency gains but some other baselines may better inform whether this adaptive control based on thresholding the approximation error is necessary - please see Questions.

References:

[1] Kunstner, Frederik, et al. "Heavy-tailed class imbalance and why adam outperforms gradient descent on language models." Advances in Neural Information Processing Systems 37 (2024): 30106-30148.

[2] Zhao, Rosie, et al. "Deconstructing what makes a good optimizer for language models." arXiv preprint arXiv:2407.07972 (2024).

Minor:
* Authors should add a reference to Appendix C after presenting Lemma 1 and Proposition 1 in the main paper.

---

> ### Author Rebuttal · Authors · 2025-07-31
>
> Thank you for your comments and raising insightful questions. We address the weaknesses and questions below. We unfortunately cannot share figures for the new experimental results due to changes in rules this year. However, we will add all new results to the paper.
>
>
> > The empirical setting is limited to two architectures on a single dataset and it would’ve been particularly interesting to see how the eigenbasis frequency results in eg. Figure 4 change for other domains, relating to previous work similarly identifying key network parameters requiring adaptivity due to issues like class imbalance for the last layer for instance [1,2].
>
> We consider three additional dataset-architecture pairs covering different modalities like graph classification in the experiments presented in Table 1+3 (ImageNet ViT, FastMRI, and OGBG workloads from the AlgoPerf benchmark). However, we agree that our analysis is currently **limited to a single model in Figure 4**.
>
> To address this limitation, we conduct a **similar analysis for the ConvNeXt V2** model trained on the Imagewoof dataset. We observe a similar trend: eigenbases for 1D parameters are updated most frequently, followed by eigenbases for $L_t$, lastly followed by eigenbases for $R_t$, which are updated least frequently. This order is consistent across at least these two architectures on the same dataset.
>
> To additionally expand beyond the vision modality and cover the mentioned class imbalance, we train a **Llama 3 model with 160M parameters on 3.2B tokens from the C4 dataset** with $\tau=0.01$ and conduct a similar analysis. We compute mean and standard errors across all (single) Kronecker factors $A_t$ for RMSNorm parameters and all $L_t$ and $R_t$ Kronecker factors for linear layers and blocks of the embedding (there are no bias parameters) at every iteration. Consistent with both experiments in the vision setting, $L_t$ is updated more often than $R_t$. On average, $L_t$ is updated almost every iteration, whereas $R_t$ is updated closer to every other iteration. The Kronecker factors $A_t$ for the normalization layers are first updated at a similar frequency to $L_t$, but, **in contrast to the vision setting**, the update frequency decreases after the first ~25% of iterations until it is closer to, but still higher than, the update frequency for $R_t$ at the end. The standard error for $A_t$ is also larger than for $L_t$ and $R_t$.
>
>
> > Authors should add a reference to Appendix C after presenting Lemma 1 and Proposition 1 in the main paper.
>
> Agreed. We have added this missing reference, thanks for pointing this out.
>
>
> > The authors’ proposed adaptive QR algorithm provides efficiency gains but some other baselines may better inform whether this adaptive control based on thresholding the approximation error is necessary - please see Questions.
>
> > How would the authors’ proposed adaptive criterion of the QR algorithm compare to having frequency warmup (starting with frequency = 1 and roughly gradual increase to F = 50) on all of the parameters, or a fraction of the parameters (in terms of loss vs wall-clock time)?
>
> Thank you for the suggestion. While you are correct that we could schedule $F$, one could also **consider a schedule for $\tau$**. We argue that this is the right comparison. Therefore, your question raises an important issue: whether scheduling $F$ or scheduling $\tau$ is more useful for controlling efficiency.
>
> We argue that $\tau$ enables more fundamental control. Specifically:
> - Setting the frequency $F$ fixes the computational cost (i.e., number of eigendecompositions) across all factor matrices and implicitly determines how the approximation quality $\tau$ varies for each Kronecker factor.
> - In contrast, setting the threshold $\tau$ directly controls the approximation quality and implicitly leads to a varying frequency $F$ schedule for each Kronecker factor.
>
> In general, we believe that controlling inexactness via the approximation quality threshold $\tau$, rather than the cost (frequency of eigendecomposition $F$), better captures its potential impact on Shampoo's convergence. For instance, in the extreme case where a Kronecker factor’s eigenbasis does not change at all, $F$ provides no mechanism to skip unnecessary computations, resulting in wasted cost without convergence benefits.
>
> To empirically validate this intuition, we conducted an **additional experiment** on the Imagewoof ViT workload using a simple two-phase schedule for $F$ and $\tau$:
> - We set $F=1$ for the first 10% of iterations and $F=1000$ for the remaining 90%.
> - We set $\tau=0.01$ for the first 10% of iterations and $\tau=0.8$ for the remaining 90%.
>
> Both schedules yielded comparable final loss. However, the $\tau$-based schedule resulted in significantly fewer total eigendecompositions and thus reduced overall wall-clock time.
>
> We will include a subsection in the paper discussing these results in more detail.
>
>
> > Do the authors have more intuition about the observed need for frequent updating for layer norm parameters? If the LayerNorms did not have learnable parameters, would the matrix layers then require more frequent computations (the LayerNorm parameters act as some output scale for the given layer)?
>
> Thank you for the question. We do not have a satisfactory explanation for this observation.
>
> We also **retrained the ViT without learnable LayerNorm parameters on Imagewoof** and created the same plot from Figure 4 (right). Interestingly, removing the learnable LayerNorm parameters does not lead to any significant difference in the number of eigendecompositons for $L_t$ and $R_t$.

---

> > ### Comment · Reviewer_aEP3 · 2025-08-04
> >
> > I thank the authors for their response and interesting follow-up results. They have addressed my comments and I am willing to raise my score to 5, accept.

---

> > > ### Author Response · Authors · 2025-08-09
> > >
> > > We are glad you found the follow-up results interesting and thank you for adjusting your score.

---

### Decision · Program_Chairs · 2025-09-17

**Decision:**

Accept (spotlight)

**Comment:**

This paper investigates the Shampoo optimizer, with a particular focus on learning rate grafting and the eigenbasis update frequency, two tricks that are seen as crucial for strong empirical performance. The authors find that grafting addresses the problem of staleness of eigenvalues incurred by infrequent updates. The authors further propose an adaptive update criterion based on approximation error, which allows different components of a model to refresh at different rates, and they demonstrate efficiency gains without significant loss of performance. The reviewers found the paper to be well-written and clear, offering insightful connections between Shampoo heuristics and variants (such as SOAP). The empirical examination of eigenbasis update frequency and the adaptive approach, was appreciated. Weaknesses included the initially narrow empirical scope, the looseness of some theoretical bounds and the concern that the adaptive threshold still introduces a tunable hyperparameter. Rebuttal directly addressed these issues by presenting new results on ConvNeXt V2, ViT on Imagewoof, a Llama model on C4 and clarifying that threshold-based control offers more fundamental advantages than fixed-frequency schedules.

Finally all reviewers recommended accept and I also find the paper to be a solid, insightful contribution that will be of clear interest to the optimization and deep learning community, and acceptance is strongly justified.